# ENERGY-BASED AUTOMATED MODEL EVALUATION

**Ru Peng**[1]* **Heming Zou**[1]* **Haobo Wang**[1] **Yawen Zeng**[2] **Zenan Huang**[1] **Junbo Zhao**[1]†

[1]Zhejiang University    [2]ByteDance

{rupeng,zouheming,wanghaobo,lccurious,j.zhao}@zju.edu.cn
yawenzeng11@gmail.com

## ABSTRACT

The conventional evaluation protocols on machine learning models rely heavily on a labeled, i.i.d-assumed testing dataset, which is not often present in real-world applications. The Automated Model Evaluation (AutoEval) shows an alternative to this traditional workflow, by forming a proximal prediction pipeline of the testing performance without the presence of ground-truth labels. Despite its recent successes, the AutoEval frameworks still suffer from an overconfidence issue, substantial storage and computational cost. In that regard, we propose a novel measure — **M**eta-**D**istribution **E**nergy **(MDE)** that allows the AutoEval framework to be both more efficient and effective. The core of the MDE is to establish a *meta-distribution* statistic, on the information (energy) associated with individual samples, then offer a smoother representation enabled by energy-based learning. We further provide our theoretical insights by connecting the MDE with the classification loss. We provide extensive experiments across modalities, datasets and different architectural backbones to validate MDE's validity, together with its superiority compared with prior approaches. We also prove MDE's versatility by showing its seamless integration with large-scale models, and easy adaption to learning scenarios with noisy- or imbalanced- labels. Code and data are available: https://github.com/pengr/Energy_AutoEval

## 1 INTRODUCTION

Model evaluation grows critical in research and practice along with the tremendous advances of machine learning techniques. To do that, the standard evaluation is to evaluate a model on a pre-split test set that is i)-fully labeled; ii)-drawn i.i.d. from the training set. However, this conventional way may fail in real-world scenarios, where there often encounter distribution shifts and the absence of ground-truth labels. In those environments with distribution shifts, the performance of a trained model may vary significantly (Quinonero-Candela et al., 2008; Koh et al., 2021b), thereby limiting in-distribution accuracy as a weak indicator of the model's generalization performance. Moreover, traditional cross-validation (Arlot & Celisse, 2010) and annotating samples are both laborious tasks, rendering it impractical to split or label every test set in the wild. To address these challenges, predicting a model's performance on various out-of-distribution datasets without labeling, a.k.a *Automated Model Evaluation* (AutoEval), has emerged as a promising solution and received some attention (Deng et al., 2021; Guillory et al., 2021; Garg et al., 2022).

The AutoEval works are typically dedicated to the characteristics of the model's output on data. The past vanilla approaches are developed to utilize the model confidence on the shifted dataset (Guillory et al., 2021; Garg et al., 2022), and they have evidently suffered from the overconfidence problem. As a result, some other metric branches are spawned, such as the agreement score of multiple models' predictions (Chen et al., 2021a; Jiang et al., 2021), the statistics (e.g. distributional discrepancy) of network parameters (Yu et al., 2022; Martin et al., 2021). Deng et al. (2021); Peng et al. (2023) introduce the accuracy of auxiliary self-supervised tasks as a proxy to estimate the classification accuracy. The computational and/or storage expense is deemed as another problem in these AutoEval methods. For instance, Deng & Zheng (2021) propose to measure the distributional differences between training and an out-of-distribution (OOD) testing set. Despite the feasibility of such an approach, it demands to access the training set in every iterative loop of evaluation. While these prior approaches indeed prove the validity of AutoEval, most (if not all) of them involve extra heavy

---

*Equal contribution.
†Corresponding author.

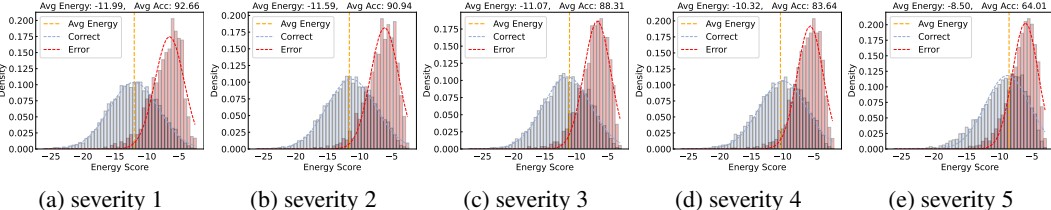

Figure 1: Trends between average energy and classification accuracy over different severity levels, we take CIFAR-10-C fog sets as an example. The density (*y-axis*) is calculated as the proportion of classified correctly (*blue unimodal*) / incorrectly (*red unimodal*) data within different energy ranges to the total samples. As the severity of the dataset strengthens, the accuracy degrades while the average energy increases accordingly (*i.e., the yellow dash line is moving to the right*).

compute and/or external storage cost – including the training set being stored/indexed, (retrained) model parameters, separate self-training objective – which may cause unneglectable overhead to the system. To that regard, we pose the motivation of this work: **can we establish a simpler, but more efficient and effective AutoEval framework, without resorting to much external resources?**

To reach this goal is challenging. Most importantly, we hope to re-establish the AutoEval workflow by associating the inherent characteristic of the networks' output with its input more directly and transparently. Profoundly, we utilize *energy*, as introduced by LeCun et al. (2006) in the Energy-based Model (EBM), that we find aligned with our purpose. In this context, "energy" denotes the scalar value assigned to a data point as fitted into the data manifold through the hypothesis class. In essence, the classifier can be viewed as an EBM (Zhao et al., 2016; Grathwohl et al., 2019) with a notable nature – "*the correct classified data are given low energies, and vice versa*". Based on this finding, we empirically probe the relationship between energy and accuracy in Fig. 1. We observe a similar phenomenon with the previous AutoEval studies: as the dataset shift intensifies, the accuracy degrades while the average energy increases accordingly.

In line with the above observations, we propose a novel measure – **M**eta-**D**istribution **E**nergy **(MDE)** for accuracy prediction. Specifically, we present MDE as a meta-distribution (hence the name) statistic that is normalized based on characterizing the information (energy) of each sample individually. This indicator transforms the information quantity of overall samples into a statistic of the probability distribution, providing a softer representation of the dataset's distribution compared to the initial energy score. Also, we provide theoretical analysis for our method by connecting MDE to classification loss through Theorem 3.1. This theoretical justification indicates that under mild assumptions, the MDE measure consistently correlates with negative log-likelihood loss, thus reflecting the trends in model generalization. Hence, we posit a hypothesis as follows: the MDE – calculated from the test set alone – provides insights into the prediction of the model's testing accuracy.

For the measures derived in this way, we conduct rigorous empirical studies on different datasets — guided by the theory we pose above — that we prove the MDE on the test sets strongly correlates with its performance (*Spearman's rank correlation $\rho$ for vision $>0.981$ and for text $>0.846$*). These results experimentally substantiate that our MDE's capability to predict the model's OOD test accuracy. Thus far, as a holistic AutoEval pipeline, we wish to emphasize that MDE outperforms the prior training-free AutoEval approaches, and is more memory- and compute-efficient than the training-must methods. It is further capable to serve as a plug-and-play module to elegantly evaluate off-the-shelf models including large-scale ones. Under varied cross-modal, data, and backbone setups, MDE significantly surpasses its prior counterpart and sets a new SOTA record for test performance evaluation. Further, we show that MDE remains effective even in strongly noisy and class-imbalanced scenarios. Finally, we visualize some in-depth analysis to demonstrate the interpretability of our method. In summary, we list our contributions as follows: (i)-we propose a simple but effective, plug-and-play AutoEval pipeline, which broadens the AutoEval technique towards production in the real world. (ii)-MDE sets a new SOTA benchmark by significantly exceeding existing works and is backed by theoretical insights for its effectiveness.

## 2 RELATED WORKS

**Automated Model Evaluation** is proposed to evaluate model performance on previously unseen, unlabeled datasets, hence also called unsupervised accuracy estimation. Recent methods mainly

consider exploiting the properties of model output on unlabeled datasets for evaluation. Preliminary research focuses on confidence score (Guillory et al., 2021; Garg et al., 2022; Lu et al., 2023c; Wang et al., 2023), such as softmax probability. Subsequently, a variety of directions have emerged along with this research field consistently develops: disagreement of multiple models' predictions (Madani et al., 2004; Donmez et al., 2010; Platanios et al., 2016; 2017; Chen et al., 2021a; Jiang et al., 2021; Baek et al., 2022), distribution discrepancy (Sun et al., 2021; Yu et al., 2022; Deng & Zheng, 2021), norm and power law of network parameters (Unterthiner et al., 2020; Martin et al., 2021; Jain et al., 2023), decomposition values of prediction matrix (Jaffe et al., 2015; Deng et al., 2023), bucketing based on decision boundaries (Hu et al., 2023; Xie et al., 2023; Tu et al., 2023; Miao et al., 2023), conditional independence assumptions (Steinhardt & Liang, 2016). In addition,Deng et al. (2021; 2022); Peng et al. (2023) add self-supervised tasks as a surrogate measure to estimate the classifier's accuracy. Chen et al. (2021b) proposed an importance weighting approach guided by priori knowledge in accuracy estimation, akin to the re-weighting in Zhang et al. (2020). Chen et al. (2022) propose SEES to estimate performance shift in both label and feature distributions. Meanwhile, a useful testbed was proposed to evaluate the model's generalization ability (Sun et al., 2023). Encouragingly, the AutoEval concept has been extened to broader domains, e.g. database (Schelter et al., 2020), structured data (Maggio et al., 2022), autonomous driving (Guan & Yuan, 2023), text classification (Elsahar & Gallé, 2019), feature engineering (Li et al., 2023), even the most closely watched LLM (Yue et al., 2023) and AIGC (Lu et al., 2023a). Our approach differs from these above studies but aims to present a *solid* paradigm to address this evaluation task more perfectly.

**Predicting ID Generalization Gap** is to predict the performance gap between the paired training-test set, thereby facilitating an understanding of the model's generalization capability to in-distribution data. This field has explored a long line of work from complexity measurement on training models and training data, representative studies involve Neyshabur et al. (2017); Dziugaite & Roy (2017); Arora et al. (2018); Zhou et al. (2018); Jiang et al. (2018; 2019); Nagarajan & Kolter (2019a;b); Corneanu et al. (2020); Zhang et al. (2021). For example, Jiang et al. (2018) introduces an indicator of layer-wise margin distribution for the generalization prediction. Corneanu et al. (2020) derives a set of persistent topology measures to estimate the generalization gap. Chuang et al. (2020) gauges the generalization error via domain-invariant representations. Baldock et al. (2021) propose a measure of example difficulty (*i.e.,* prediction depth) in the context of deep model learning. The above works are developed for the same distribution between the training and test sets without accessing test data. In contrast, we focus on predicting model accuracy across *various* OOD datasets using the *attributes* of the test sample.

**Energy-based Model** is a non-normalized probabilistic model that captures dependencies between variables by associating scalar energy to each variable (LeCun et al., 2006). EBMs do not impose restrictions on the tractability of normalized constants, making them more flexible for parameterization. As a result, researchers have started using EBMs to model more expressive families of probability distributions (Ranzato et al., 2006; 2007). However, the unknown normalization constant of EBMs makes training particularly difficult. Hence, Xie et al. (2016) first uses Langevin dynamics to effectively train a CNN classifier that can be regarded as an EBM. Follow-up works investigate training EBMs through Markov chain Monte Carlo (MCMC) techniques (Du & Mordatch, 2019; Song & Kingma, 2021). After (Xie et al., 2016; Grathwohl et al., 2019) revealed that the classifier essentially acts as an energy model, energy-based applications have sprung up, such as GAN (Zhao et al., 2016), video (Xie et al., 2019), point cloud (Xie et al., 2021), voxel (Xie et al., 2018), trajectory (Xu et al., 2022) and molecules (Liu et al., 2021). With the support of the discovery that "*the correct classified data has higher energy, and vice versa*", energy view has also been applied to OOD detection (Liu et al., 2020). But unlike Liu et al. (2020) using energy to detect OOD test samples that are different from training distributions, our work is towards predicting the model's accuracy on unlabeled OOD test sets. Inspired by these pioneering works, we formulate energy-driven statistics as the *accuracy surrogate* to assess the feasibility of our method in AutoEval task.

## 3   ENERGY-BASED AUTOMATED MODEL EVALUATION

In this section, we propose an energy-based AutoEval framework pivoted on the *meta-distribution energy*. First, we formulate the AutoEval problem (Section 3.1). We then describe the meta-distribution energy in detail. (Section 3.2). Also, we connect the meta-distribution energy to a mathematical theorem on classification loss for a theoretical guarantee of our method (Section 3.3). A pseudo-code is provided in algorithm 1.

---

**Algorithm 1** Automated Model Evaluation via Meta-distribution Energy

---

**Input:** Synthetic sets $\left\{\mathcal{D}_s^i\right\}_{i=1}^C$, unlabeled OOD set $\mathcal{D}_u$, classifier $f$, energy function $Z(x; f)$.
1: **for** $i = 1, 2, ..., C$ **do**
2:      $acc_i =: \mathbb{E}_{(x,y) \sim \mathcal{D}_s^i} [\mathbb{I}[y \neq \arg\max_{j \in \mathcal{Y}} \text{Softmax}(f_j(x))]]$
3:      $\text{MDE}_i =: -\frac{1}{|N|} \sum_{i=1}^N \log \text{Softmax} Z(x; f)$
4: **end for**
5: Fit a linear regressor $(w, b)$ from the collection of $\{(acc_i, \text{MDE}_i)\}_{i=1}^C$
6: Regress the accuracy of $f$ on $\mathcal{D}_u$: $a\hat{c}c_u =: \mathbb{E}_{x \sim \mathcal{D}_u} [w^T \text{MDE} + b]$
7: Mean absolute error: $\varepsilon = |acc_u - a\hat{c}c_u|$
**Output:** Correlation coefficients $R^2$, $r$, $\rho$ and mean absolute error $\varepsilon$.

---

### 3.1 PROBLEM STATEMENT

**Notations.** In this work, we consider a multi-class classification task with input space $\mathcal{X} \subseteq \mathbb{R}^d$ and label space $\mathcal{Y} = \{1, \ldots, K\}$. We denote $\mathcal{P}_\mathcal{S}$ and $\mathcal{P}_\mathcal{T}$ as the source and target distributions over $\mathcal{X} \times \mathcal{Y}$, respectively. Let $p_S$ and $p_T$ define the corresponding probability density functions. Given a training dataset $\mathcal{D}_o^S$ i.i.d sampled from $\mathcal{P}_\mathcal{S}$, we train a probabilistic classifier $f : \mathbb{R}^d \to \Delta_K$, where $\Delta_K$ is a unit simplex in $K$-1 dimensions. For a held-out test set $\mathcal{D}_t^S = \{(\boldsymbol{x}_i^s, y_i^s)\}_{i=1}^M$ drawn from $\mathcal{P}_\mathcal{S}$, when accessed at data point $(\boldsymbol{x}^s, y^s)$, $f$ returns $\hat{y} =: \arg\max_{j \in \mathcal{Y}} \text{softmax}(f_j(\boldsymbol{x}^s))$ as the predicted label and $f_j(\boldsymbol{x}^s)$ as the associated logits of $j$-th class. Given the label $y^s$, the classification error (i.e., 0-1 loss) on that sample is computed as $\mathcal{E}(f(\boldsymbol{x}^s), y^s) := \mathbb{I}[y^s \neq \hat{y}]$. By calculating the errors on all points of $\mathcal{D}_t^S$, we can determine the in-distribution test accuracy of classifier $f$ on $\mathcal{P}_\mathcal{S}$.

**Automated Model Evaluation**. However, under distribution shift ($p_S \neq p_T$), the in-distribution (source) test accuracy of $\mathcal{D}_t^S$ fails to actually reflect the generalization performance of $f$ on target $p_T$. To this end, this work aims to evaluate how well $f$ performs on the varied out-of-distribution (target) data without access to labels. Specifically, given a trained $f$ and an unlabeled OOD dataset $\mathcal{D}_u^T = \{(\boldsymbol{x}_i^t)\}_{i=1}^N$ with $N$ samples drawn i.i.d. from $p_T$, we expect to develop a quantity that strongly correlated to the accuracy of $f$ on $\mathcal{D}_u^T$. Note that, the target distribution $p_T$ has the same $K$ classes as the source distribution $p_S$ (known as the closed-set setting) in this work. And unlike domain adaptation, our goal is not to adapt the model to the target data.

### 3.2 META-DISTRIBUTION ENERGY FOR AUTOEVAL

In this part, we elaborate on the MDE measure and the AutoEval pipeline centered on it.

**Meta-Distribution Energy.** The energy-based model (EBM) (LeCun et al., 2006) was introduced to map each data point $x$ to a scalar dubbed as *energy* via an energy function $Z(x) : \mathbb{R}^D \to \mathbb{R}$. It could transform the energy values into a probability density $p(x)$ through the Gibbs distribution:

$$p(y \mid x) = \frac{e^{-Z(x,y)/T}}{\int_{y'} e^{-Z(x,y')/T}} = \frac{e^{-Z(x,y)/T}}{e^{-Z(x)/T}}, \tag{1}$$

where the denominator $\int_{y'} e^{-Z(x,y')/T}$ is the partition function by marginalizing over $y$, and $T$ is the positive temperature constant. Now the negative of the log partition function can express the *Gibbs free energy* $Z(x)$ at the data point $x$ as:

$$Z(x) = -T \cdot \log \int_{y'} e^{-Z(x,y')/T}. \tag{2}$$

In essence, the energy-based model has an inherent connection with the discriminative model. To interpret this, we consider the above-mentioned discriminative classifier $f : \mathbb{R}^d \to \Delta_K$, which maps a data point $x \in \mathbb{R}^d$ to $K$ real number known as logits. These logits are used to parameterize a categorical distribution using the Softmax function:

$$p(y \mid x) = \frac{e^{f_y(x)/T}}{\sum_{j=1}^K e^{f_j(x)/T}}, \tag{3}$$

where $f_y(x)$ denotes the $y$-th term of $f(x)$, i.e., the logit corresponding to the $y$-th class. Combining Eq. 2 and Eq. 3, we can derive the energy for a given input data $(x, y)$ as $Z(x, y) = -f_y(x)$. Thus, given a neural classifier $f(x)$ and an input point $x \in \mathbb{R}^D$, we can express the energy function with respect to the denominator of the Softmax function:

$$Z(x; f) = -T \cdot \log \sum_{j=1}^{K} e^{f_j(x)/T}. \tag{4}$$

Assume an unlabeled dataset $\mathcal{D}_{\mathrm{u}} = \{(x_i)\}_{i=1}^{N}$ with $N$ samples, we define MDE as a meta-distribution statistic re-normalized on the energy density $Z(x; f)$ of every data point $x$:

$$\mathrm{MDE}(x; f) = -\frac{1}{|N|} \sum_{i=1}^{N} \log \mathrm{Softmax}\, Z(x; f) = -\frac{1}{|N|} \sum_{i=1}^{N} \log \frac{e^{Z(x_n; f)}}{\sum_{i=1}^{N} e^{Z(x_i; f)}}, \tag{5}$$

where $Z(x_n; f)$ indicates the free energy of $n$-th data point $x_n$, $|N|$ is the cardinality of $\mathcal{D}_{\mathrm{u}}^{T}$. This indicator transforms global sample information into a meta-probabilistic distribution measure. Aiding by the information normalization, MDE offers a smoother dataset representation than initial energy score *AvgEnergy* (also proposed by us) which solely averages the energy function on the dataset.

**AutoEval Pipeline.** We give the procedure for using MDE to predict the OOD testing accuracy, but other measurements are also applicable. Given a model $f$ to be evaluated, we first compute the value pairs of its true accuracy and MDE on the synthetic test set. Then, the accuracy of the OOD test set can be estimated by a simple linear regression. Consequently, we write down the forms as follows:

$$acc =: \mathbb{E}_{(x,y) \sim \mathcal{D}} \left[ \mathbb{I} \left[ y \neq \arg\max_{j \in \mathcal{Y}} \mathrm{Softmax}\, (f_j(x)) \right] \right], \tag{6}$$

$$\hat{acc} =: \mathbb{E}_{x \sim \mathcal{D}} \left[ w^T \mathrm{MDE}(x; f) + b \right], \tag{7}$$

$$\varepsilon = |acc - \hat{acc}|, \tag{8}$$

where $acc$ and $\hat{acc}$ are the ground-truth and estimated accuracy, respectively. $\mathbb{I}[\cdot]$ is an indicator function, $(x, y)$ is input data and class label, $\varepsilon$ is the mean absolute error for accuracy estimation.

**Remarks.** According to our formulation, our MDE method poses three appealing properties: *i)*-a *training-free approach* with high efficiency by dispensing with extra overhead; *ii)*-a *calibration* with built-in temperature scaling to get rid of the overconfidence issue of only using model logits; *iii)*-a re-normalized meta-distribution statistic with a *smoother* dataset representation. These properties largely guarantee the efficiency and effectiveness of our MDE algorithm. More interestingly, since our method smoothly condenses all logits, our MDE metric demonstrates excellent robustness to label bias and noise; see Section 4.5 for details.

### 3.3 THEORETICAL ANALYSIS

From a theoretical side, we first set an assumption that the sample energy needs to be satisfied when the discriminative classifier (i.e., an EBM) minimizes the loss function:

**Corollary 3.1** *For a sample $(x, y)$, incorrect answer $\bar{y}$ and positive margin $m$. Minimizing the loss function $\mathcal{L}$ will satisfy $Z(x, y; f) < Z(x, \bar{y}; f) - m$ if there exists at least one point $(z_1, z_2)$ with $z_1 + m < z_2$ such that for all points $(z_1', z_2')$ with $z_1' + m \geq z_2'$, we have $\mathcal{L}_{[Z_y]}(z_1, z_2) < \mathcal{L}_{[Z_y]}(z_1', z_2')$, where $[Z_y]$ contains the vector of energies for all values of $y$ except $y$ and $\bar{y}$.*

**Theorem 3.1** *Given a well-trained model $f$ with optimal loss $\mathcal{L}_{nll}$, for each sample point $(x_i, y_i)$, the difference between its classification risk and MDE can be characterized as follows:*

$$\Delta^i = \mathrm{MDE}^i - \mathcal{L}_{nll}^i = f_{y_i}(x_i)/T - \max_{j \in \mathcal{Y}} f_j(x_i)/T = \begin{cases} 0, & \text{if } j = y_i, \\ < 0, & \text{if } j \neq y_i, \end{cases} \tag{9}$$

*where $Y$ is the label space, $\mathrm{MDE}$ is the proposed meta-distribution energy indicator, $\mathcal{L}_{nll}$ is the negative log-likelihood loss, $T$ is the temperature constant approximate to 0.*

We can ascertain whether label $y_i$ corresponds to the maximum logits by comparing the term Eq. 9 and zero, thus assessing the model's accuracy. Thus, we theoretically establish the connection between MDE and accuracy by a mathematical theorem. Detailed theoretical analysis in Appendix C.

Table 1: Correlation comparison with existing methods on synthetic shifted datasets of CIFAR-10, CIFAR-100, and TinyImageNet, MNLI. We report coefficient of determination ($R^2$) and Spearson's rank correlation ($\rho$) (higher is better). The training-must methods marked with "*", while the others are training-free methods. The highest score in each row is highlighted in **bold**.

| Dataset | Network | ConfScore | | Entropy | | Frechet | | ATC | | AgreeScore* | | ProjNorm* | | MDE | |
|---|---|---|---|---|---|---|---|---|---|---|---|---|---|---|---|
| | | $\rho$ | $R^2$ | $\rho$ | $R^2$ | $\rho$ | $R^2$ | $\rho$ | $R^2$ | $\rho$ | $R^2$ | $\rho$ | $R^2$ | $\rho$ | $R^2$ |
| CIFAR-10 | ResNet-20 | 0.991 | 0.953 | 0.990 | 0.958 | 0.984 | 0.930 | 0.962 | 0.890 | 0.990 | 0.955 | 0.974 | 0.954 | **0.992** | **0.964** |
| | RepVGG-A0 | 0.979 | 0.954 | 0.981 | 0.946 | 0.982 | 0.864 | 0.959 | 0.888 | 0.981 | 0.950 | 0.970 | 0.969 | **0.985** | **0.980** |
| | VGG-11 | 0.986 | 0.956 | 0.989 | 0.960 | 0.990 | 0.908 | 0.947 | 0.907 | 0.989 | 0.903 | 0.985 | 0.955 | **0.991** | **0.974** |
| | Average | 0.985 | 0.954 | 0.987 | 0.955 | 0.985 | 0.901 | 0.956 | 0.895 | 0.987 | 0.936 | 0.976 | 0.959 | **0.989** | **0.973** |
| CIFAR-100 | ResNet-20 | 0.962 | 0.906 | 0.943 | 0.870 | 0.964 | 0.880 | 0.968 | 0.923 | 0.970 | 0.925 | 0.967 | 0.927 | **0.981** | **0.961** |
| | RepVGG-A0 | 0.985 | 0.938 | 0.977 | 0.926 | 0.955 | 0.864 | 0.982 | 0.963 | 0.983 | 0.953 | 0.973 | 0.933 | **0.992** | **0.978** |
| | VGG-11 | 0.979 | 0.950 | 0.972 | 0.937 | 0.986 | 0.889 | **0.991** | 0.958 | 0.980 | 0.953 | 0.966 | 0.881 | **0.991** | **0.960** |
| | Average | 0.975 | 0.931 | 0.964 | 0.911 | 0.968 | 0.878 | 0.980 | 0.948 | 0.978 | 0.944 | 0.969 | 0.914 | **0.988** | **0.966** |
| TinyImageNet | ResNet-50 | 0.932 | 0.711 | 0.937 | 0.755 | 0.957 | 0.818 | 0.986 | 0.910 | 0.971 | 0.895 | 0.944 | 0.930 | **0.994** | **0.971** |
| | DenseNet-161 | 0.964 | 0.821 | 0.925 | 0.704 | 0.948 | 0.813 | 0.989 | 0.943 | 0.983 | 0.866 | 0.957 | 0.930 | **0.994** | **0.983** |
| | Average | 0.948 | 0.766 | 0.931 | 0.730 | 0.953 | 0.816 | 0.988 | 0.927 | 0.977 | 0.881 | 0.950 | 0.930 | **0.994** | **0.977** |
| MNLI | BERT | 0.650 | 0.527 | 0.790 | 0.536 | 0.517 | 0.479 | 0.650 | 0.487 | 0.608 | 0.457 | 0.636 | 0.547 | **0.853** | **0.644** |
| | RoBERTa | 0.734 | 0.470 | 0.741 | 0.516 | 0.587 | 0.494 | 0.643 | 0.430 | 0.825 | 0.682 | 0.790 | 0.531 | **0.846** | **0.716** |
| | Average | 0.692 | 0.499 | 0.766 | 0.526 | 0.552 | 0.487 | 0.647 | 0.459 | 0.717 | 0.570 | 0.713 | 0.539 | **0.850** | **0.680** |

# 4 EXPERIMENTS

In this chapter, we assess the MDE algorithm across various data setups in both visual and text domains, which includes: correlation studies, accuracy prediction errors, the hyper-parameter sensitivity, as well as two stress tests: strong noise and class imbalance.

## 4.1 EXPERIMENTAL SETUP

In this work, we evaluate each method on the image classification tasks CIFAR-10, CIFAR-100 (Krizhevsky et al., 2009), TinyImageNet (Le & Yang, 2015), ImageNet-1K (Deng et al., 2009), WILDS (Koh et al., 2021a) and the text inference task MNLI (Williams et al., 2018). See Appendix A for details.

**Training Details.** Following the practice in Deng et al. (2023), we train models using a public implementations[1] for CIFAR datasets. The models in ImagNet-1K are provided directly by timm library (Wightman et al., 2019). Likewise, we use the commonly-used scripts[2] to train the models for TinyImageNet. Similarly, for the WILDS data setup, we align with the methodology proposed by (Garg et al., 2022) for the selection and fine-tuning of models. For the MNLI setup, we use the same training settings as (Yu et al., 2022).

**Compared Baselines.** We consider **eight** methods as compared baselines: 1) *Average Confidence (ConfScore)* (Hendrycks & Gimpel, 2016), 2) *Average Negative Entropy (Entropy)* (Guillory et al., 2021), 3) *Frechet Distance (Frechet)* (Deng & Zheng, 2021), 4) *Agreement Score (AgreeScore)* (Jiang et al., 2021), 5) *Average Thresholded Confidence (ATC)* (Garg et al., 2022), 6) *Confidence Optimal Transport (COT)* (Lu et al., 2023b), 7) *Average Energy (AvgEnergy)*, 8) *Projection Norm (ProjNorm)* (Yu et al., 2022), 9) *Nuclear Norm (NuclearNorm)* (Deng et al., 2023). The first six existing methods are developed using the model's output. The AvgEnergy we devised is based on initial energy score, highly tied to our MDE. The final two are currently the SOTA methods. For further details, see Appendix B.

**Evalutaion Metrics.** To evaluate the performance of accuracy prediction, we use coefficients of determination ($R^2$), Pearson's correlation ($r$), and Spearman's rank correlation ($\rho$) (higher is better) to quantify the correlation between measures and accuracy. Also, we report the mean absolute error (MAE) results between predicted accuracy and ground-truth accuracy on the naturally shifted sets.

---

[1]https://github.com/chenyaofo/pytorch-cifar-models
[2]https://github.com/pytorch/vision/tree/main/references/classification

Table 2: Correlation comparison with SOTA and highly related methods on synthetic shifted datasets of different data setup.

Table 3: Mean absolute error (MAE) comparison with SOTA and highly related methods on natural shifted datasets of different data setup.

| Dataset | Network | NuclearNorm | | AvgEnergy | | MDE | |
|---|---|---|---|---|---|---|---|
| | | $\rho$ | $R^2$ | $\rho$ | $R^2$ | $\rho$ | $R^2$ |
| CIFAR-10 | ResNet-20 | **0.996** | 0.959 | 0.989 | 0.955 | 0.992 | **0.964** |
| | RepVGG-A0 | 0.989 | 0.936 | **0.990** | 0.959 | 0.985 | **0.980** |
| | VGG-11 | **0.997** | 0.910 | 0.993 | 0.957 | 0.991 | **0.974** |
| | Average | **0.994** | 0.935 | 0.991 | 0.957 | 0.989 | **0.973** |
| CIFAR-100 | ResNet-20 | **0.986** | 0.955 | 0.977 | 0.956 | 0.981 | **0.961** |
| | RepVGG-A0 | **0.997** | 0.949 | 0.986 | 0.968 | 0.992 | **0.978** |
| | VGG-11 | **0.997** | 0.947 | 0.986 | 0.964 | 0.991 | **0.960** |
| | Average | **0.993** | 0.950 | 0.983 | 0.963 | 0.988 | **0.966** |
| TinyImageNet | ResNet-50 | 0.991 | 0.969 | 0.991 | 0.966 | **0.994** | **0.971** |
| | DenseNet-161 | 0.993 | 0.968 | 0.983 | 0.961 | **0.994** | **0.983** |
| | Average | 0.992 | 0.969 | 0.987 | 0.964 | **0.994** | **0.977** |
| MNLI | BERT | 0.650 | 0.521 | 0.783 | 0.539 | **0.853** | **0.644** |
| | RoBERTa | 0.685 | 0.471 | 0.832 | 0.650 | **0.846** | **0.716** |
| | Average | 0.668 | 0.496 | 0.808 | 0.595 | **0.850** | **0.680** |

| Dataset | Natural Shifted Sets | NuclearNorm | AvgEnergy | MDE |
|---|---|---|---|---|
| CIFAR-10 | CIFAR-10.1 | 1.53 | 1.55 | **0.86** |
| | CIFAR-10.2 | 2.66 | 1.47 | **1.01** |
| | CINIC-10 | 2.95 | 2.63 | **0.48** |
| | STL-10 | 6.54 | 5.86 | **4.78** |
| | Average | 3.42 | 2.88 | **1.78** |
| TinyImageNet | TinyImageNet-V2-A | 1.59 | 0.80 | **0.54** |
| | TinyImageNet-V2-B | 2.36 | 1.92 | **1.11** |
| | TinyImageNet-V2-C | 1.91 | 1.76 | **0.88** |
| | TinyImageNet-S | 1.90 | 1.24 | **0.47** |
| | TinyImageNet-R | 3.96 | 2.72 | **2.41** |
| | TinyImageNet-Vid | 9.16 | 8.49 | **6.08** |
| | TinyImageNet-Adv | 6.01 | 5.66 | **3.59** |
| | Average | 3.84 | 3.23 | **2.15** |
| MNLI | QNLI | 7.82 | 6.30 | **5.56** |
| | RTE | 6.49 | 5.39 | **3.96** |
| | WNLI | 8.69 | 7.50 | **6.06** |
| | SciTail | 7.21 | 6.48 | **4.79** |
| | ANLI | 12.01 | 10.48 | **8.42** |
| | Average | 8.44 | 7.23 | **5.76** |

Table 4: Mean absolute error (MAE) comparison with existing methods on natural shifted datasets of CIFAR-10, TinyImageNet, and MNLI. The training-must methods marked with "*", while the others are training-free methods. The best result in each row is highlighted in **bold**.

| Dataset | Unseen Test Sets | ConfScore | Entropy | Frechet | ATC | AgreeScore* | ProjNorm* | MDE |
|---|---|---|---|---|---|---|---|---|
| CIFAR-10 | CIFAR-10.1 | 9.61 | 3.72 | 5.55 | 4.87 | 3.37 | 2.65 | **0.86** |
| | CIFAR-10.2 | 7.12 | 8.95 | 6.70 | 5.90 | 3.78 | 4.59 | **1.01** |
| | CINIC-10 | 7.24 | 8.16 | 9.81 | 5.91 | 4.62 | 9.43 | **0.48** |
| | STL-10 | 10.45 | 15.25 | 11.80 | 15.92 | 11.77 | 12.98 | **4.78** |
| | Average | 8.61 | 9.02 | 8.47 | 8.15 | 5.89 | 7.41 | **1.78** |
| TinyImageNet | TinyImageNet-V2-A | 7.22 | 5.67 | 7.68 | 4.78 | 4.37 | 3.77 | **0.54** |
| | TinyImageNet-V2-B | 8.80 | 10.61 | 11.65 | 5.57 | 6.36 | 5.04 | **1.11** |
| | TinyImageNet-V2-C | 10.67 | 8.04 | 14.58 | 9.38 | 5.69 | 3.56 | **0.88** |
| | TinyImageNet-S | 11.44 | 9.54 | 8.32 | 13.17 | 6.35 | 9.80 | **0.47** |
| | TinyImageNet-R | 10.18 | 8.02 | 11.28 | 14.81 | 7.10 | 9.50 | **2.41** |
| | TinyImageNet-Vid | 13.12 | 15.36 | 13.57 | 16.20 | 19.72 | 10.11 | **6.08** |
| | TinyImageNet-Adv | 14.85 | 14.93 | 10.27 | 15.66 | 10.98 | 12.94 | **3.59** |
| | Average | 10.90 | 10.31 | 11.05 | 11.37 | 8.65 | 7.82 | **2.15** |
| MNLI | QNLI | 16.10 | 17.31 | 15.57 | 10.54 | 14.33 | 15.88 | **5.56** |
| | RTE | 12.32 | 18.18 | 16.39 | 14.46 | 10.92 | 9.43 | **3.96** |
| | WNLI | 9.99 | 17.37 | 21.67 | 21.10 | 15.15 | 15.78 | **6.06** |
| | SciTail | 16.85 | 17.27 | 16.56 | 11.88 | 9.06 | 9.97 | **4.79** |
| | ANLI | 25.14 | 22.19 | 14.69 | 20.85 | 12.34 | 17.93 | **8.42** |
| | Average | 16.08 | 18.46 | 16.98 | 15.77 | 12.36 | 13.80 | **5.76** |

## 4.2 Main Results: Correlation Study

We summarize the correlation results ($R^2$ and $\rho$) for all methods under different settings in Table 1, 2, 6 and Fig. 2. Encouragingly, our MDE surpasses every (even SOTA) baseline in a fair comparison across modalities, datasets, and backbones. We discuss these results from the following aspects:

In Table 1 and 6, MDE significantly outperforms common **training-free** methods. Specifically, the average $R^2$ of MDE on CIFAR-10 (0.973), CIFAR-100 (0.966), TinyImageNet (0.977), ImageNet (0.960) and MNLI (0.680) exceeds the ConfScore, Entropy, Frechet, and ATC by a notable margin. These gains may benefit from the temperature scaling in MDE re-calibrating confidences. The MDE is also superior to the **training-must** AgreeScore and ProjNorm. This advantaged scheme improves performance, reduces cost, and seamlessly meets the evaluation needs of the popular LLM.

**MDE vs. SOTA and Highly related methods.** As shown in Table 2, MDE achieves better performance than the recent SOTA NuclearNorm in almost all setups, especially in the MNLI setup. This series of results substantiates that MDE is a competitive technique with extensive applicability.

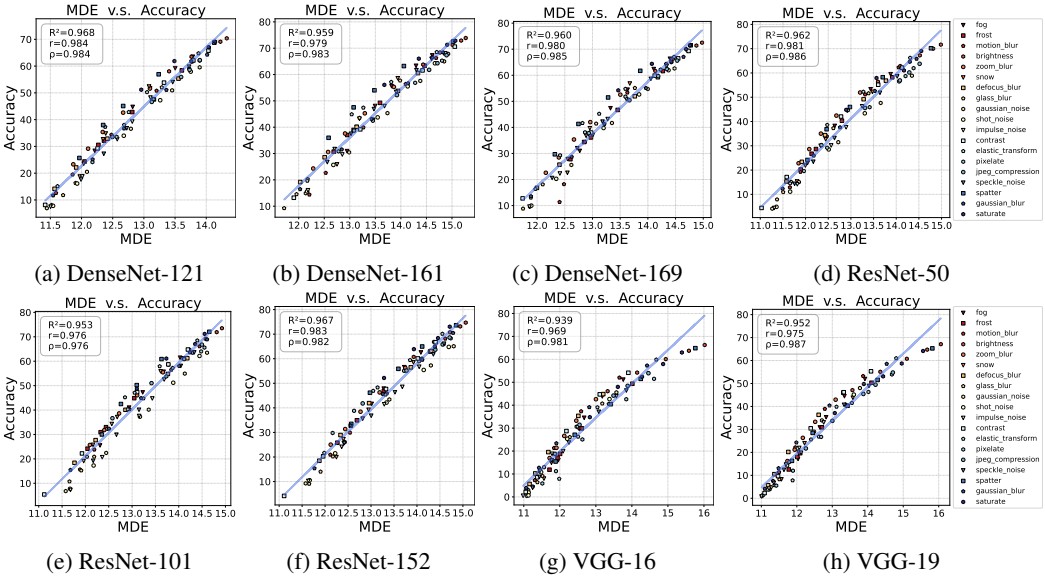

Figure 2: MDE's coefficients of determination ($R^2$), Pearson's correlation ($r$) and Spearman's rank correlation ($\rho$) on synthetic shifted datasets of ImageNet setup.

Notably, MDE consistently outperforms the well-performing AvgEnergy which is highly tied to us. It confirms the energy-based indicator can strongly correlate to accuracy. More importantly, MDE yields a stronger correlation by a smoother measure after re-normalizing the global sample energies.

**Bigger and Textual datasets: ImageNet-1K, MNLI.** Further, we present scatter plots of MDE on ImageNet-1k in Fig. 2. We emphasize that MDE remains robustly linearly related to the performance of off-the-shelf models, even in the extreme case of test accuracy below 20 (see subplots (a) and (g)). On the textual MNLI dataset, the average correlation obtained by our MDE is also effective ($R^2$=0.680, $\rho$=0.850). These findings greatly bolster the deployment of our approach across diverse real-world scenarios. The complete set of scatter plots can be found in Appendix H.

## 4.3 MAIN RESULTS: ACCURACY PREDICTION ERROR

We show the mean absolute error (MAE) results for all methods in predicting accuracy on real-world datasets in Table 3, 4, 7 and 8. For each natural shifted set, we report its MAE value as the average across all backbones. Among seven datasets, we conclude that our method reduces the average MAE from 5.25 to 3.14 by about **40.0%** upon prior SOTA method (NuclearNorm), thus setting a new SOTA benchmark in accuracy prediction. Further, MDE shows strong performance regardless of the classification domain (e.g. MNLI) or the classification granularity (ranging from CIFAR-10 to TinyImageNet). Interestingly, in certain extremely hard test sets (e.g. STL-10, TinyImagenet-Adv, ANLI), other methods fail with a relatively poor estimated error while we perform well yet. These results are not only excellent, and robust but also substantiated by the optimal correlation observed between MDE and accuracy. This reminds us that the AutoEval technique heavily relies on the correlation degree between measure and accuracy.

## 4.4 ANALYSIS OF HYPERPARAMETER SENSITIVITY

As we adopt the MDE-based AutoEval framework, we want to know the sensitivity of its performance to hyperparameters. So we study the impact of variations in temperature and random seed on performance. Here, we report the results using VGG-11 on the CIFAR-10 setup, which remains consistent in the subsequent experiments unless otherwise stated. All results of this section are placed in the appendix.

**Scaled temperature constants.** As an important factor in the MDE calculation, we study the temperature constant $T$ from 0.01 to 100. As figure 7 (a) shows, the performance declines when the

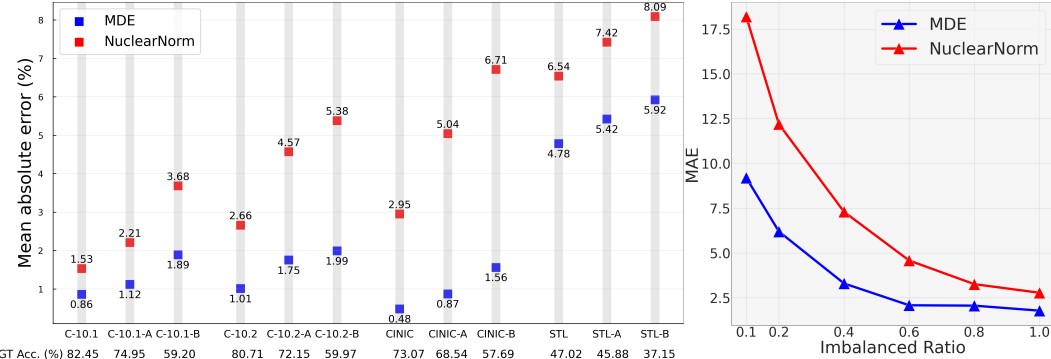

Figure 3: Mean absolute errors on two stress tests: *(left)*-strongly noisy and *(right)*-class imbalance.

temperature increases and the best performance appears in $T = 1$. The correlation coefficients and MAE of a broader range of temperature can be found in Table 9.

**Different random seeds.** To examine if the experimental results are robust to the initial random state, we pick different random seeds for training (use 1 as the default seed). As figure 7 (b) shows, the performance of our framework is robust to randomness.

### 4.5 STRESS TESTS: STRONGLY NOISY AND CLASS IMBALANCED CASES

**Strongly Noisy.** In the previous analysis, we tested our method on the naturally shifted test sets. Considering that real-world scenarios may be more complex, we test the robustness of MDE and NuclearNorm (SOTA) in a "more realistic" test environment by applying new transformations on naturally shifted test sets. Note that new transformations are strictly screened to have no overlap with various transformations in the synthetic set (i.e. CIFAR-10-C). Specifically, we use Cutout (DeVries & Taylor, 2017), Shear, Equalize and ColorTemperature (Cubuk et al., 2019) to generate CIFAR-10.1-A/B, CIFAR-10.2-A/B, CINIC-10-A/B, STL-10.1-A/B. We note the following observations from the left of Fig. 3. First, the greater the shifted intensity, the harder both methods are to predict accuracy. The accuracy prediction results in the re-transformed test sets (-A/B) are worse than the untransformed state. Also, CINIC-10 and STL-10 with larger shifts, experience more performance decline compared to other datasets. Second, under the noised data undergoing new transformations, our method consistently achieves more superior results (MAE $< 5.92$) than NuclearNorm.

**Class Imbalance.** Considering that real-world data is usually not class-balanced like our work, some classes are under-sampled or over-sampled, resulting in label shift ($p_S(y) \neq p_T(y)$). To study the effect of class imbalance, we create long-tail imbalanced test sets from synthetic datasets (CIFAR-10-C with the 2-th severity level). Specifically, we apply exponential decay (Cao et al., 2019) to control the proportion of different classes. It is represented by the imbalance ratio ($r$) – the ratio between sample sizes of the least frequent and most frequent classes – that ranges from $\{0.1, 0.2, 0.4, 0.6, 0.8, 1.0\}$. As shown in the right of Fig. 3, our method is robust under moderate imbalance ($r \geq 0.4$) than NuclearNorm. Certainly, when there is a severe class imbalance ($r \leq 0.2$), our method is also seriously affected by label shift, but it still precede NuclearNorm. At this time, considering extra techniques such as label shift estimation (Lipton et al., 2018) may be a potential idea for addressing this issue.

## 5 CONCLUSION

In this work, we introduce a novel measure, the Meta-Distribution Energy (MDE), to enhance the efficiency and effectiveness of the AutoEval framework. Our MDE addresses the challenges of overconfidence, high storage requirements, and computational costs by establishing the MDE – a meta-distribution statistic on the energy of individual samples, which is supported by theoretical theorems. Through extensive experiments across various modalities, datasets, and architectural backbones, we demonstrate the superior performance and versatility of MDE via micro-level results, hyper-parameter sensitivity, stress tests, and in-depth visualization analyses.

## ACKNOWLEDGEMENTS

This work is majorly supported by the National Key Research and Development Program of China (No. 2022YFB3304100), Fundamental Research Funds for the Central Universities (Project QiZhen @ZJU), and in part by the NSFC Grants (No. 62206247)

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

Table 5: Details of the datasets considered in our work.

| Train (Source) | Valid (Source) | Evaluation (Target) |
|---|---|---|
| CIFAR-10 (train) | CIFAR-10 (valid) | 95 CIFAR-10-C datasets, CIFAR-10.1, CIFAR-10.2, CINIC-10 |
| CIFAR-100 (train) | CIFAR-100 (valid) | 95 CIFAR-100-C datasets |
| ImageNet-1K (train) | ImageNet-1K (valid) | 95 ImageNet-C datasets, 3 ImageNet-v2 datasets, ImageNet-Sketch, ImageNet-Rendition, ImageNet-Adversarial, ImageNet-VidRobust |
| TinyImageNet (train) | TinyImageNet (valid) | 75 TinyImageNet-C datasets, 3 TinyImageNet-v2 datasets, TinyImageNet-Sketch, TinyImageNet-Rendition, TinyImageNet-Adversarial, TinyImageNet-VidRobust |
| MNLI (train) | MNLI (valid) | MNLI-M, MNLI-MM, SNLI, BREAK-NLI, HANS, SNLI-HRAD, 4 STRESS-TEST datasets, SICK, EQUATE, QNLI, RTE, WNLI, SciTail, ANLI |
| FMoW (2002-12) (train) | FMoW(2002-12) (valid) | FMoW (2013-15, 2016-17) X (All, Africa, Americas, Oceania, Asia, and Europe) |
| RxRx1 (train) | RxRx1(id-val) | RxRx1 (id-test, OOD-val, OOD-test) |
| Camelyon17 (train) | Camelyon17(id-val) | Camelyon17 (id-test, OOD-val, OOD-test) |

## A    DETAILS OF DATASET SETUP

In our work, we consider both natural and synthetic distribution shifts in empirical evaluation. A summary of the datasets we used is shown in Table 5. We elaborate on the setting of datasets and models as follows:

**CIFAR-10.** (i)-Model. We use ResNet-20 (He et al., 2016), RepVGG-A0 (Ding et al., 2021), and VGG-11 (Simonyan & Zisserman, 2014). They are trained from scratch using the CIFAR-10 training set (Krizhevsky et al., 2009). (ii)-Synthetic Shift. We use CIFAR-10-C benchmark (Hendrycks & Dietterich, 2019) to study the synthetic distribution shift. The CIFAR-10-C datasets have controllable corruption with 95 sub-datasets, involving 19 types of corruption with 5 different intensity levels applied to the CIFAR-10 validation set. (iii)-Natural Shift. These include three test sets: 1) CIFAR-10.1 and CIFAR-10.2 (Recht et al., 2018) with reproduction shift, 2) CINIC-10 (Darlow et al., 2018) that is a selection of downsampled 32x32 ImageNet images for CIFAR-10 class labels.

**CIFAR-100.** The datasets and models used are the same as the CIFAR-10 setup, but here we only consider the case of synthetic shift, i.e. CIFAR-100-C (Hendrycks & Dietterich, 2019).

**ImageNet-1K.** (i)-Model. We use the image models provided by timm library (Wightman et al., 2019). They comprise three series of representative convolution neural networks: DenseNet (DenseNet-121/161/169/202) (Huang et al., 2017), ResNet (ResNet-50/101/152), VGG (VGG-16/19). These models are either trained or fine-tuned on ImageNet training set (Deng et al., 2009). (ii)-Synthetic Shift. Similar to CIFAR10-C, we employ the ImageNet-C (Hendrycks & Dietterich, 2019) to investigate the synthetic shift. This dataset spans 19 types of corruption with 5 severity levels. (iii)-Natural Shift. We consider five natural shifts, which involve: 1) dataset reproduction shift for ImageNet-V2-A/B/C (Recht et al., 2019), 2) sketch shift for ImageNet-S(ketch) (Wang et al., 2019), 3) style shift for ImageNet-R(endition) with 200 ImageNet classes (Hendrycks et al., 2021a), 4) adversarial shift for ImageNet-Adv(ersarial) with 200 ImageNet classes (Hendrycks et al., 2021b), 5) temporal shift for ImageNet-Vid(Robust) with 30 ImageNet classes (Shankar et al., 2021).

**TinyImageNet.** (i)-Model. We use 2 classic classifiers: DenseNet-161 and ResNet-50. They are pre-trained on ImageNet and fine-tuned on the TinyImageNet training set (Le & Yang, 2015). (ii)-Synthetic Shift. Following the practice of ImageNet-C, we adopt TinyImageNet-C which only applies 15 types of corruptions with 5 intensity levels to the TinyImageNet validation set. (iii)-Natural Shift. We select the same naturally shifted dataset as in the ImageNet-1K Setup, but only pick the parts that share the same classes as TinyImageNet.

**MNLI.** (i)-Model. For the natural language inference task, we utilize pre-trained versions of BERT (Devlin et al., 2018) and RoBERTa (Liu et al., 2019) from HuggingFace library (Wolf et al., 2020). These transformer models are fine-tuned on the MNLI training set (Williams et al., 2018). (ii)-Synthetic Shift. In the MNLI setup, we combine these datasets to examine synthetic shift: MNLI-M, MNLI-MM, SNLI (Bowman et al., 2015), BREAK-NLI (Glockner et al., 2018), HANS (McCoy et al., 2019), SNLI-HRAD (Gururangan et al., 2018), STRESS-TEST (Naik et al., 2018), SICK (Marelli et al., 2014) and EQUATE (Ravichander et al., 2019). The STRESS-TEST containing 4 sub-datasets, includes shifts such as length mismatch, spelling errors, word overlap, and antonyms. (more detailed descriptions of these datasets can be found in Zhou et al. (2020)) (iii)-Natural Shift. We discuss two types of shifts: (1) domain shift in QNLI, RTE, WNLI (Wang et al., 2018), SciTail (Khot et al., 2018); (2) adversarial shift in ANLI (Nie et al., 2019);

**Camelyon17-WILD.** (i)-Model. Following the setting of (Garg et al., 2022), we use ResNet-50 and DenseNet-121. They are pre-trained on ImageNet and fine-tuned on Camelyon17's training set. (ii)-Synthetic and Natural Shift. We used the official synthetic and natural shifted datasets provided in (Koh et al., 2021a).

**RxRx1-WILD.** (i)-Model. Following the setting of (Garg et al., 2022), we use ResNet-50 and DenseNet-121. They are pretrained on ImageNet and fine-tuned on RxRx1's training set. (ii)-Synthetic and Natural Shift. We used the official synthetic and natural shifted datasets provided in (Koh et al., 2021a).

**FMoW-WILD.** (i)-Model. Following the setting of (Garg et al., 2022), we use ResNet-50 and DenseNet-121. They are pretrained on ImageNet and fine-tuned on FMoW's training set. (ii)-Synthetic and Natural Shift. Similarly, we obtain 12 different synthetic and natural shifted datasets by considering images in different years and by considering five geographical regions as subpopulations (Africa, Americas, Oceania, Asia, and Europe) separately and together according to (Koh et al., 2021a).

## B  BASELINE METHODS

Below we briefly present the baselines compared in our work, where we denote the classifier $f$, and the unlabeled dataset $D_u$ drawn from target distribution $\mathcal{P}_{\mathcal{T}}$:

**Average Confidence (ConfScore).** The model's accuracy on target data is estimated as the expected value of the maximum softmax confidence (Hendrycks & Gimpel, 2016):

$$\text{ConfScore} = \mathbb{E}_{x \sim \mathcal{D}_u} \left[ \max_{j \in \mathcal{Y}} \text{Softmax}(f_j(x)) \right]. \tag{10}$$

**Average Negative Entropy (Entropy).** The target accuracy of a model is predicted by the expected value of the negative entropy (Guillory et al., 2021):

$$\text{Entropy} = \mathbb{E}_{x \sim \mathcal{D}_u} \left[ \text{Ent}_{j \in \mathcal{Y}} \text{Softmax}(f_j(x)) \right], \tag{11}$$

where $\text{Ent}(p) = -p \cdot \log(p)$. Note that, the *Difference of Confidence (DoC)* (Guillory et al., 2021) – equals to the ConfScore and Entropy indicators – when there is no label space shift between source and target distributions, i.e. closed-set setting.

**Frechet Distance (Frechet).** The model's accuracy on target can be assessed by the Frechet Distance between the features of the training set $\mathcal{D}_o$ and the target set (Deng & Zheng, 2021):

$$\text{Frechet} = \text{FD}\left(\mathbb{E}_{x \sim \mathcal{D}_o}\left[(f(x)], \mathbb{E}_{x \sim \mathcal{D}_u}\left[(f(x)]\right), \tag{12}$$

where $\text{FD}(\mathcal{D}_o, \mathcal{D}_u) = \|\mu_o - \mu_u\|_2^2 + \text{Tr}\left(\Sigma_o + \Sigma_u - 2(\Sigma_o \Sigma_u)^{\frac{1}{2}}\right)$, $\mu$ and $\Sigma$ are the mean feature vectors and the covariance matrices of a dataset.

**Agreement Score (AgreeScore).** The model accuracy is estimated as the expected disagreement of two models (trained on the same training set but with different randomization) on target data (Jiang et al., 2021):

$$\text{AgreeScore} = \mathbb{E}_{x \sim \mathcal{D}_u} \left[ \mathbb{I} \left[ \max_{j \in \mathcal{Y}} \text{Softmax}(f_j^1(x)) \neq \max_{j \in \mathcal{Y}} \text{Softmax}(f_j^2(x)) \right] \right] \tag{13}$$

where $f^1$ and $f^2$ are two models that are trained on the same training set but with different initializations.

**Average Thresholded Confidence (ATC).** This method learns a threshold $t$ on model confidence scores from source validation data $\mathcal{D}_t$, then predicts the target accuracy as the proportion of unlabeled target data with a score higher than the threshold (Garg et al., 2022):

$$\text{ATC} = \mathbb{E}_{x \sim \mathcal{D}_u} \left[ \mathbb{I} \left[ \underset{j \in \mathcal{Y}}{\text{Ent}} \, \text{Softmax}(f_j(x)) < t \right] \right], \tag{14}$$

$$\mathbb{E}_{x \sim \mathcal{D}_t} \left[ \mathbb{I} \left[ \underset{j \in \mathcal{Y}}{\text{Ent}} \, \text{Softmax}(f_j(x)) < t \right] \right] = \mathbb{E}_{(x,y) \sim \mathcal{D}_t} \left[ \mathbb{I} \left[ y \neq \arg \max_{j \in \mathcal{Y}} \text{softmax} \, (f_j(x)) \right] \right]. \tag{15}$$

**Average Energy (AvgEnergy).** This measure is a self-designed metric closely tied to our MDE, which predicts the model's accuracy by the expected value of the energy on target data:

$$\text{AvgEnergy} = \mathbb{E}_{x \sim \mathcal{D}_u} \left[ Z(x; f) \right] = \mathbb{E}_{x \sim \mathcal{D}_u} \left[ -T \cdot \log \sum_{j=1}^{K} e^{f_j(x)/T} \right]. \tag{16}$$

**Projection Norm (ProjNorm).** This algorithm pseudo-labels the target samples using the classifier $f$, then uses these pseudo data $(x, \widetilde{y})$ to train a new model $\widetilde{f}$ from the initialized network $f_0$. The model's target accuracy is predicted by the parameters difference of two model (Yu et al., 2022):

$$\widetilde{y} =: \arg \max_{j \in \mathcal{Y}} \text{softmax} \, (f_j(\boldsymbol{x}^s)) \tag{17}$$

$$\text{ProjNorm} = \left\| \theta_f - \theta_{\widetilde{f}} \right\|_2. \tag{18}$$

**Nuclear Norm (NuclearNorm).** This approach uses the normalized value of the nuclear norm (i.e., the sum of singular values) of the prediction matrix to measure the classifier accuracy on the target dataset (Deng et al., 2023):

$$\text{NuclearNorm} = \mathbb{E}_{x \sim \mathcal{D}_u} \left[ \frac{\| \text{Softmax}(f_j(x)) \|_*}{\sqrt{\min (|N|, K) \cdot |N|}} \right]. \tag{19}$$

where $\|p\|_*$ is the nuclear norm of $p$, and $|N|$ is the cardinality of $\mathcal{D}_u$, $K$ is the number of classes.

**Confidence Optimal Transport (COT).** This approach leverages the optimal transport framework to predict the error of a model as the Wasserstein distance between the predicted target class probabilities and the true source label distribution (Lu et al., 2023b):

$$\text{COT} = W_\infty \left( f_\# \mathcal{P}_{\mathcal{T}}(c), \mathcal{P}_{\mathcal{S}}(y) \right). \tag{20}$$

where $W_\infty$ is the Wasserstein distance with $c(x, y) = \|x - y\|_\infty$, $c(x, y)$ is a cost function that tells us the cost of transporting from location $x$ to location $y$. $f_\# \mathcal{P}_{\mathcal{T}}(c)$ is defined to be the pushforward of a covariate distribution $\mathcal{P}_{\mathcal{T}}$.

## C  DETAILED THEORETICAL ANALYSIS

Recalling Theorem 3.1, we provide some more detailed discussions of this theorem here, including its basic assumptions and a complete proof of the theorem. We start with an assumption that the well-trained discriminative classifier (i.e., an EBM) makes correct inferences of sample $(x_i, y_i)$ with minimum energy.

**Assumption C.1**  $\forall y \in \mathcal{Y}$ and $y \neq y_i$, for sample $(x_i, y_i)$, the model will give the correct answer for $x_i$ if $Z(x_i, y_i; f) < Z(x_i, y; f)$.

To ensure the correct answer is robustly stable, we may opt to enforce that its energy is lower than the energies of incorrect answer $\bar{y}_i$ by a positive margin $m$. This modified assumption is as follows:

**Assumption C.2** *For a incorrect answer $\bar{y}_i$, sample $(x_i, y_i)$ and positive margin $m$, the inference algorithm will give the correct answer for $x_i$ if $Z(x_i, y_i; f) < Z(x_i, \bar{y}_i; f) - m$*

Now, we are ready to deduce the sufficient conditions for the minimum loss function. Let two points $(z_1, z_2)$ and $(z_1', z_2')$ belong to the feasible region $R$, such that $(z_1, z_2) \in HP_1$ (that is, $z_1 + m < z_2$) and $(z_1', z_2') \in HP_2$ (that is, $z_1' + m \geq z_2'$).

**Corollary C.1** *For a sample $(x_i, y_i)$ and positive margin $m$. Minimizing the loss function $\mathcal{L}$ will satisfy assumptions C.1 or C.2 if there exists at least one point $(z_1, z_2)$ with $z_1 + m < z_2$ such that for all points $(z_1', z_2')$ with $z_1' + m \geq z_2'$, we have $\mathcal{L}_{[Z_y]}(z_1, z_2) < \mathcal{L}_{[Z_y]}(z_1', z_2')$, where $[Z_y]$ contains the vector of energies for all values of $y$ except $y_i$ and $\bar{y}_i$.*

Next, with the well-trained classifier $f$, we proceed to correlate the MDE and its classification accuracy in out-of-distribution data $(x, y) \sim p_T$. The temperature $T$ is a positive constant and defaults to 1. To do this, we first express the negative log-likelihood loss for $f$ as:

$$
\begin{aligned}
\mathcal{L}_{\text{nll}} &= \mathbb{E}_{(x,y)\sim p_T}\left(-\log\frac{e^{f_y(x)/T}}{\sum_{j=1}^{K} e^{f_j(x)/T}}\right) \\
&= \mathbb{E}_{(x,y)\sim p_T}\left(-f_y(x)/T + \log\sum_{j=1}^{K} e^{f_j(x)}\right).
\end{aligned}
\tag{21}
$$

Then, we represent the MDE computed by $f$ on $x \sim p_T$ as follows:

$$
\begin{aligned}
\text{MDE} &= \mathbb{E}_{x\sim p_T}\left(-\log\text{Softmax}\left(-T\cdot\log\sum_{j=1}^{K} e^{f_j(x)/T}\right)\right) \\
&= \mathbb{E}_{x\sim p_T}\left(-\log\frac{e^{-\log\sum_{j=1}^{K} e^{f_j(x_i)/T}}}{\sum_{i=1}^{N} e^{-\log\sum_{j=1}^{K} e^{f_j(x_i)/T}}}\right) \\
&= \mathbb{E}_{x\sim p_T}\left(\log\sum_{j=1}^{K} e^{f_j(x_i)/T} + \log\sum_{i=1}^{N}\left(\sum_{j=1}^{K} e^{f_j(x_i)/T}\right)^{-1}\right)
\end{aligned}
\tag{22}
$$

Afterward, we represent the difference between the MDE indicator and the negative log-likelihood loss as follows:

$$
\Delta = \text{MDE} - \mathcal{L}_{\text{nll}} = \mathbb{E}_{x\sim p_T}\left(\log\sum_{i=1}^{N}\left(\sum_{j=1}^{K} e^{f_j(x_i)/T}\right)^{-1}\right) + \mathbb{E}_{(x,y)\sim p_T}\left(f_y(x)/T\right)
\tag{23}
$$

For each sample point $(x_i, y_i)$, the subtraction form can be rewritten as:

$$
\begin{aligned}
\Delta^i = \text{MDE}^i - \mathcal{L}_{\text{nll}}^i &= -\log\left(\sum_{j=1}^{K} e^{f_j(x_i)/T}\right) + f_{y_i}(x_i)/T \\
&\stackrel{\lim T\to 0}{=} f_{y_i}(x_i)/T - \max_{j\in\mathcal{Y}} f_j(x_i)/T
\end{aligned}
\tag{24}
$$

$$
= \begin{cases} 0, & \text{if } j = y_i, \\ < 0, & \text{if } j \neq y_i, \end{cases}
\tag{25}
$$

which is our deduced result. In this proof, we assume an ideal situation where $T$ approaches 0, but usually in practical applications $T$ defaults to 1. Finally, by judging whether the term of Eq. 24 is less than 0, we can know whether the index $j$ corresponding to the maximum logits is the label $y$, i.e., we can obtain the accuracy value of the classifier $f$. Thus, we theoretically substantiated a correlation between MDE and accuracy by a mathematical theorem.

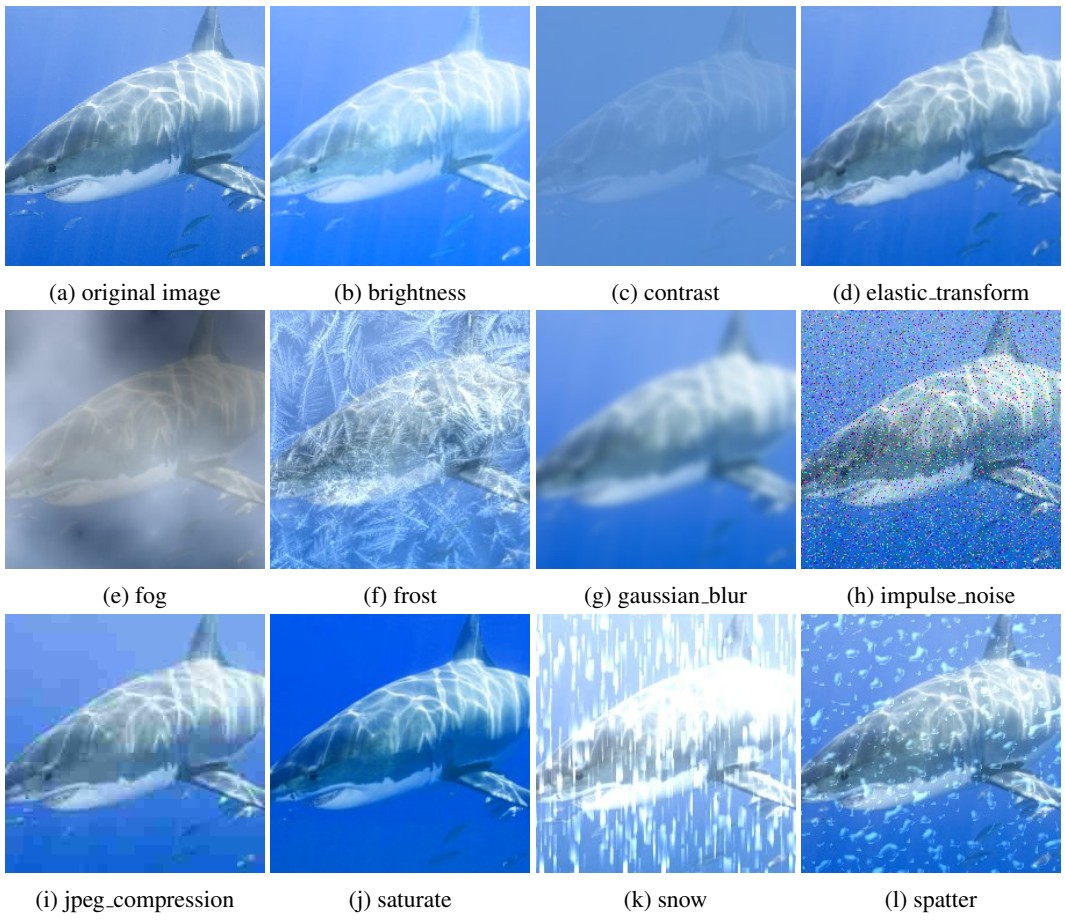

| (a) original image | (b) brightness | (c) contrast | (d) elastic_transform |
| (e) fog | (f) frost | (g) gaussian_blur | (h) impulse_noise |
| (i) jpeg_compression | (j) saturate | (k) snow | (l) spatter |

Figure 4: Visualized examples of synthetic sets in ImageNet-1k Setup

## D  Sample Visualization of Synthetic Sets

In Fig. 4, we provide some visualized examples of synthetic sets undergoing various transformations in ImageNet-1k Setup.

## E  Discussion: Class-level Correlation

From the previous results, we have observed a strong linear correlation built at the dataset level between MDE and classification accuracy. A question naturally arises: can their correlation at the category level also be established? To this end, we plot the T-SNE visualization of clustering the penultimate classifier features in Fig. 5. Whether on the ID or any OOD datasets, we found that the accuracy of each class showed a consistent trend of "co-varying increments and decrements" with its MDE, i.e., characterized by a positive linear correlation. Also, the CINIC-10 and STL-10 clusters are poorer than the remaining datasets. It is not hard to see that models can have better clustering effects because they have better classification accuracy, which corresponds to the lower accuracy of CINIC-10 and STL-10. Of course, that is because their samples are mainly transformed from the ImageNet dataset.

## F  Discussion: Meta-disrtibuton Energy from different layers

Here, we wish to explore whether features from other layers can compute MDE scores that are as discriminative as the (default) classification head features. In Fig. 6, we display the MDE scores

Table 6: Correlation comparison with existing methods on synthetic shifted dataset of ImageNet. We report coefficient of determination ($R^2$) and Spearson's rank correlation ($\rho$) (higher is better). The highest score in each row is highlighted in **bold**.

| Dataset | Network | ConfScore | | Entropy | | Frechet | | ATC | | COT | | NuclearNorm | | AvgEnergy | | MDE | |
|---|---|---|---|---|---|---|---|---|---|---|---|---|---|---|---|---|---|
| | | $\rho$ | $R^2$ | $\rho$ | $R^2$ | $\rho$ | $R^2$ | $\rho$ | $R^2$ | $\rho$ | $R^2$ | $\rho$ | $R^2$ | $\rho$ | $R^2$ | $\rho$ | $R^2$ |
| ImageNet | ResNet-152 | 0.980 | 0.949 | 0.979 | 0.946 | 0.945 | 0.879 | 0.980 | 0.899 | 0.968 | 0.943 | 0.979 | 0.961 | 0.980 | 0.955 | **0.982** | **0.967** |
| | DenseNet-169 | 0.983 | 0.953 | 0.981 | 0.931 | 0.942 | 0.878 | 0.982 | 0.891 | 0.963 | 0.934 | 0.981 | 0.956 | 0.984 | 0.955 | **0.985** | **0.960** |
| | VGG-19 | 0.966 | 0.933 | 0.968 | 0.909 | 0.978 | 0.910 | 0.975 | 0.886 | 0.981 | 0.926 | 0.978 | 0.949 | 0.980 | 0.950 | **0.987** | **0.952** |
| | Average | 0.976 | 0.945 | 0.976 | 0.929 | 0.955 | 0.889 | 0.979 | 0.892 | 0.971 | 0.934 | 0.979 | 0.955 | 0.981 | 0.953 | **0.985** | **0.960** |

Table 7: Mean absolute error (MAE) comparison with existing methods on natural shifted datasets of ImageNet. The best result in each row is highlighted in **bold**.

| Dataset | Unseen Test Sets | ConfScore | Entropy | Frechet | ATC | COT | NuclearNorm | AvgEnergy | MDE |
|---|---|---|---|---|---|---|---|---|---|
| ImageNet | ImageNet-V2-A | 8.76 | 9.72 | 6.36 | 6.88 | 7.04 | 4.04 | 3.58 | **2.21** |
| | ImageNet-V2-B | 8.59 | 10.20 | 9.51 | 8.82 | 9.02 | 5.20 | 4.41 | **2.35** |
| | ImageNet-V2-C | 14.92 | 10.27 | 9.12 | 8.74 | 8.85 | 5.66 | 5.01 | **4.27** |
| | ImageNet-V2-S | 7.30 | 9.50 | 10.45 | 9.81 | 8.11 | 7.77 | 6.40 | **5.31** |
| | ImageNet-V2-R | 15.41 | 13.63 | 12.06 | 12.70 | 12.87 | 11.34 | 10.64 | **7.68** |
| | ImageNet-V2-Vid | 15.18 | 14.26 | 16.53 | 14.42 | 13.49 | 12.77 | 11.19 | **8.09** |
| | ImageNet-V2-Adv | 19.09 | 18.28 | 18.37 | 19.20 | 14.51 | 13.61 | 11.43 | **8.42** |
| | Average | 12.75 | 12.27 | 11.77 | 11.51 | 10.56 | 8.63 | 7.52 | **5.48** |

Table 8: Mean absolute error (MAE) comparison with existing methods of natural shifted datasets on Wilds. The best result in each row is highlighted in **bold**.

| Dataset | Shift | ConfScore | Entropy | Frechet | ATC | COT | NuclearNorm | AvgEnergy | MDE |
|---|---|---|---|---|---|---|---|---|---|
| Camelyon17 | Natural Shift | 9.01 | 8.19 | 8.49 | 7.46 | 5.31 | 4.26 | 3.21 | **2.93** |
| RxRx1 | Natural Shift | 6.63 | 6.32 | 5.49 | 5.45 | 4.45 | 3.67 | 2.86 | **1.62** |
| FMoW | Natural Shift | 10.52 | 9.61 | 7.47 | 6.13 | 5.26 | 4.49 | 2.90 | **2.24** |

calculated for these features from different layers (i.e. the output of each block) on the ID and OOD test sets. Without exception, the MDE scores calculated by shallow features all fall within the same numerical range, and their discriminability is far inferior to the MDE of the classification head feature. This justifies that the powerful representation ability of the classification head feature is the foundation of why our MDE can work.

## G    DISCUSSION: ANALYZING CLASS DISTRIBUTION BY ENERGY BUCKETING

In this spot, we aim to understand the distribution of sample categories based on energy scores. In other words, what type of sample corresponds to what energy score? Specifically, in Fig.8, we divide the samples into different buckets as per the energy value, and then further analyze the proportion of each category in each block. We discuss these results from three aspects according to the ID and OOD data sets:

i) From Fig.8 (a) and (d), we can see that within each (same) energy score range, the class distribution in the ID data remains relatively balanced, while class distribution in the OOD data exhibits an imbalanced trend.

ii) Comparing Fig.8 (a) and (d), we can observe that across (different) energy score segments, the proportion of the same category of OOD data fluctuates more drastically than ID data, such as dog at the top 10% energy scores and horse at 10% 20% energy ranges in Fig.(d).

iii) The aforementioned two phenomena are also held under different backbones, as illustrated in other figures.

Table 9: MDE's coefficients of determination ($R^2$), Pearson's correlation ($r$), Spearman's rank correlation ($\rho$) and mean absolute errors (MAE) on scaled temperature constants.

| T | 0.01 | 0.5 | 1 | 2 | 3 | 4 | 5 | 6 | 7 | 8 | 9 |
|---|---|---|---|---|---|---|---|---|---|---|---|
| $\rho$ | 0.988 | 0.989 | 0.991 | 0.991 | 0.990 | 0.988 | 0.987 | 0.986 | 0.984 | 0.983 | 0.983 |
| $r$ | 0.989 | 0.988 | 0.987 | 0.981 | 0.974 | 0.968 | 0.964 | 0.961 | 0.960 | 0.959 | 0.959 |
| $r^2$ | 0.978 | 0.977 | 0.974 | 0.963 | 0.948 | 0.937 | 0.929 | 0.924 | 0.921 | 0.920 | 0.919 |
| MAE | 1.72 | 1.80 | 1.78 | 1.94 | 2.28 | 2.71 | 3.32 | 4.25 | 5.10 | 6.32 | 7.98 |
| T | 10 | 20 | 30 | 40 | 50 | 60 | 70 | 80 | 90 | 100 | |
| $\rho$ | 0.982 | 0.976 | 0.974 | 0.974 | 0.974 | 0.973 | 0.971 | 0.972 | 0.971 | 0.971 | |
| $r$ | 0.958 | 0.959 | 0.959 | 0.959 | 0.960 | 0.960 | 0.960 | 0.960 | 0.960 | 0.961 | |
| $r^2$ | 0.918 | 0.919 | 0.920 | 0.921 | 0.921 | 0.921 | 0.921 | 0.922 | 0.922 | 0.923 | |
| MAE | 9.50 | 9.87 | 10.05 | 10.16 | 10.22 | 10.24 | 10.25 | 10.25 | 10.24 | 10.26 | |

Table 10: MDE's coefficients of determination ($R^2$), Pearson's correlation ($r$), Spearman's rank correlation ($\rho$) and mean absolute errors (MAE) on different linear regressor.

| | RobustLinearRegression | LinearRegression |
|---|---|---|
| $\rho$ | 0.991 | 0.991 |
| $r$ | 0.987 | 0.987 |
| $r^2$ | 0.973 | 0.974 |
| MAE | 1.80 | 1.78 |

## H    COMPLETE SET OF CORRELATION SCATTER PLOTS

Here, we display the complete set of correlation scatter plots for all methods/datasets/model architectures, as follows in Fig. 9, 10, 11, 12, 13, 14, 15, 16, 17, 18.

## I    LIMITATION AND FUTURE WORK

We now briefly discuss the limitations of meta-distribution energy and future directions. Our method is grounded on an assumption that approximates the unknown test environments via data transformations applied to the synthetic sets. So one limitation is that our MDE hinges on sufficient samples and shift types to make accurate predictions on the OOD test set. It would be practical to reduce the sample requirements of this method, ideally to be a one-sample version of MDE. Another issue is that MDE sometimes performs poorly on "extreme" shifts, as the energy of the data point doesn't reflect its information anymore, under these challenging scenarios such as adversarial attacks and class imbalance. This limitation may require new tailored techniques to be addressed, which suggests an interesting avenue for future work. Furthermore, we believe the concept of Autoeval can also play a role in many more AI fields, such as text-video retrieval (Han et al., 2023), machine translation (Peng et al., 2022), sentiment analysis (Lin et al., 2023), which also represents a potential research direction in the future.

## J    DIFFERENT LINEAR REGRESSORS.

For a analysis of whether the regression performance of MDE is influenced by the choice of regression models, we selected both linear regressors and robust linear regressors for comparison. As indicated in Table 10, the results suggest that the selection of different regression models has no significant impact on the accuracy prediction performance.

## K    AUTOEVAL DIFFERENCE FROM UNCERTAINTY ESTIMATION AND OOD DETECTION

AutoEval, uncertainty estimation and out-of-distribution (OOD) detection are significantly different tasks. **First**, the three tasks have different goals. Given labeled source data and unlabeled target

data, uncertainty estimation is to estimate the confidence of model predictions to convey information about when a model's output should (or should not) be trusted, AutoEval directly predicts the accuracy of model output. Unlike OOD detection, which aims to identify outlier test samples that are different from training distributions, AutoEval is an unsupervised estimation for the model's accuracy across the entire test set. In this regard, AutoEval is a task that assesses the effectiveness and deployment worthiness of a model by directly predicting its accuracy in a testing environment. **Second**, our work was not inspired by uncertainty estimation or OOD detection techniques, but rather by the characteristics of the energy that fulfilled our desire to build an efficient and effective AutoEval framework.

## L   META-DISTRIBUTION ENERGY V.S. SOFTMAX SCORE

Here, we demonstrate that the relationship between MDE and Softmax Score is not a simple replacement by comparing them from three aspects:

i) **They are different in essence and mathematical form.** Essentially, MDE is defined as a meta-distribution statistic of non-normalized energy at the dataset level, while Softmax Score represents the maximum value of the normalized logit vector for an individual sample. In terms of mathematical formulas, they have the following distinct expressions:

$$\text{MDE}(x; f) = -\frac{1}{|N|} \sum_{i=1}^{N} \log \text{Softmax } Z(x; f) = -\frac{1}{|N|} \sum_{i=1}^{N} \log \frac{e^{Z(x_n; f)}}{\sum_{i=1}^{N} e^{Z(x_i; f)}}, \quad (26)$$

$$\max_y p(y \mid \mathbf{x}) = \max_y \frac{e^{f_y(\mathbf{x})}}{\sum_i e^{f_i(\mathbf{x})}} = \frac{e^{f^{\max}(\mathbf{x})}}{\sum_i e^{f_i(\mathbf{x})}} \quad (27)$$

ii) **Their usage in the AutoEval task is different.** Softmax score typically reflects the classification accuracy through measures such as the mean (ConfScore, Entropy), mean difference (DOC), or the data proportion below a certain threshold (ATC). On the other hand, MDE predicts accuracy by a regression model.

iii) **MDE is more suitable than Softmax score for AutoEval task.** To demonstrate this, we decompose the softmax confidence by logarithmizing it as follows:

$$\log \max_y p(y \mid \mathbf{x}) = E\left(\mathbf{x}; f(\mathbf{x}) - f^{\max}(\mathbf{x})\right)$$

$$(28)$$

$$= E(\mathbf{x}; f) + f^{\max}(\mathbf{x})$$

Then, we find $f^{max}(x)$ tends to be lower and $E(x; f)$ tends to be higher for OOD data, and vice versa. This shifting results in Softmax score that is no longer suitable for accuracy prediction, while MDE is not affected by this bothersome issue.

## M   COMPARISON IN TERMS OF EVALUATION TIME AND MEMORY USAGE BETWEEN MDE AND THE TRAINING-MUST METHOD

In this section, we want to clarify the advantage of MDE over the training-must approach in terms of time and memory consumption. However, due to the significantly different workflows of various methods (e.g., training the model from scratch, fine-tuning the model, calculating model features, statistically analyzing dataset distribution, computing the disagreement of ensemble prediction, etc.), it is impossible to compare them directly and fairly. So, we simplify this problem to compare the time complexity and space complexity of different methods in a rough granularity:

**For time complexity:**

AC = ANE<ATC<AvgEnergy = MDE (ours) = NuclearNorm<Frechet<ProjNorm(training-must) <AgreeScore (training-must).

**For space complexity:**

training-free methods (AC, ANE, ATC, AvgEnergy, MDE(ours), NuclearNorm, Frechet) <training-must methods (ProjNorm, AgreeScore).

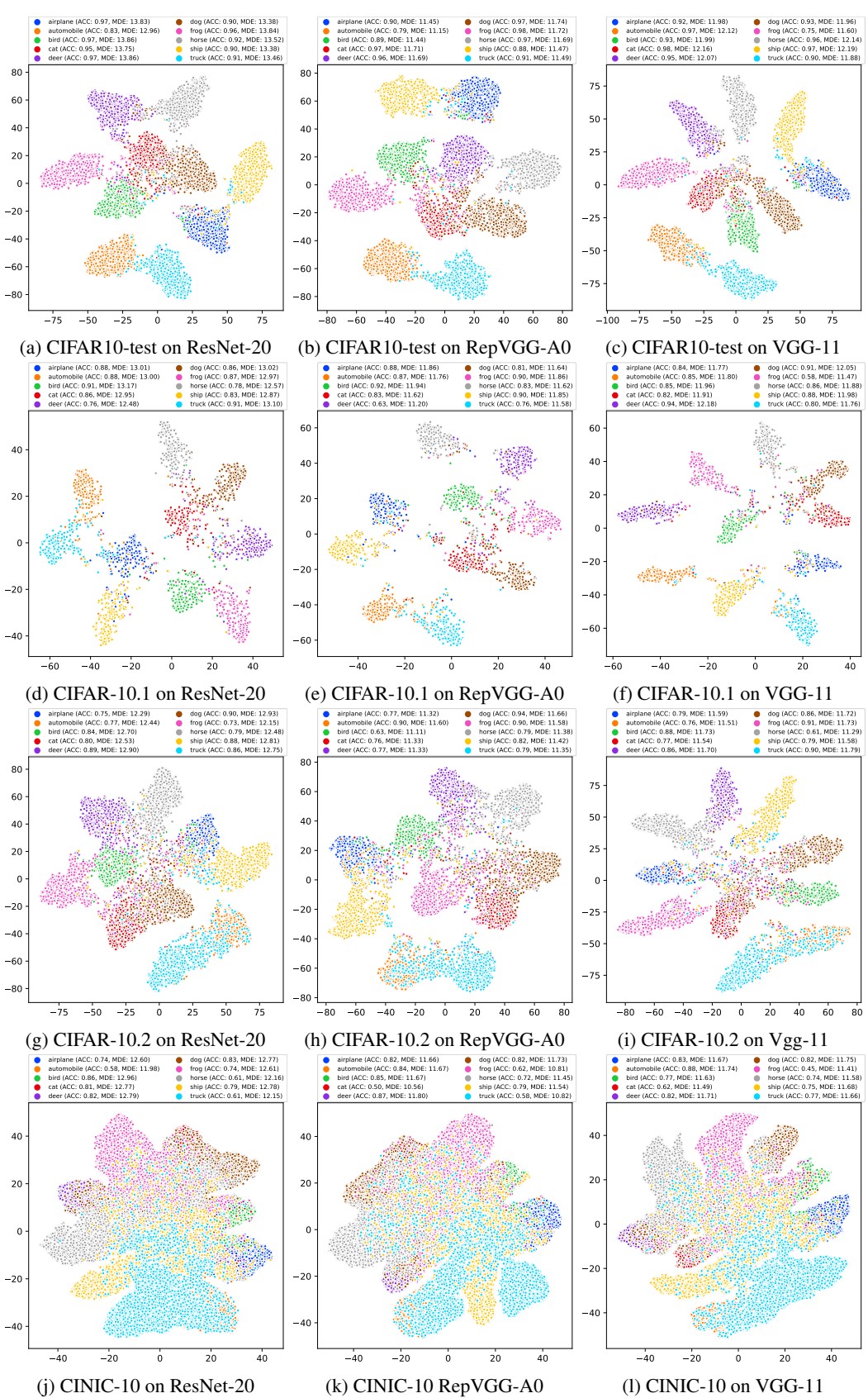

(a) CIFAR10-test on ResNet-20  (b) CIFAR10-test on RepVGG-A0  (c) CIFAR10-test on VGG-11

(d) CIFAR-10.1 on ResNet-20  (e) CIFAR-10.1 on RepVGG-A0  (f) CIFAR-10.1 on VGG-11

(g) CIFAR-10.2 on ResNet-20  (h) CIFAR-10.2 on RepVGG-A0  (i) CIFAR-10.2 on Vgg-11

(j) CINIC-10 on ResNet-20  (k) CINIC-10 RepVGG-A0  (l) CINIC-10 on VGG-11

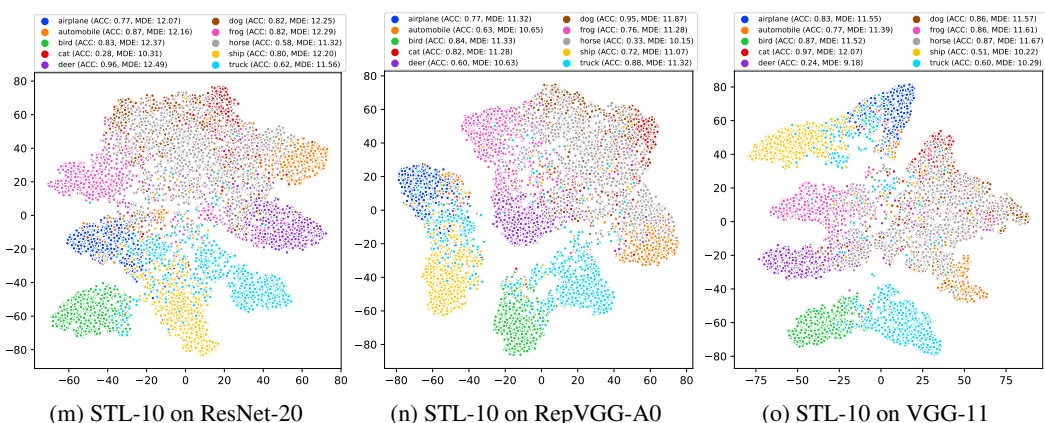

(m) STL-10 on ResNet-20    (n) STL-10 on RepVGG-A0    (o) STL-10 on VGG-11

Figure 5: T-SNE visualization of the classification features clustering on CIFAR-10 setup. Different colors correspond to different classes, and its accuracy and MDE are placed above.

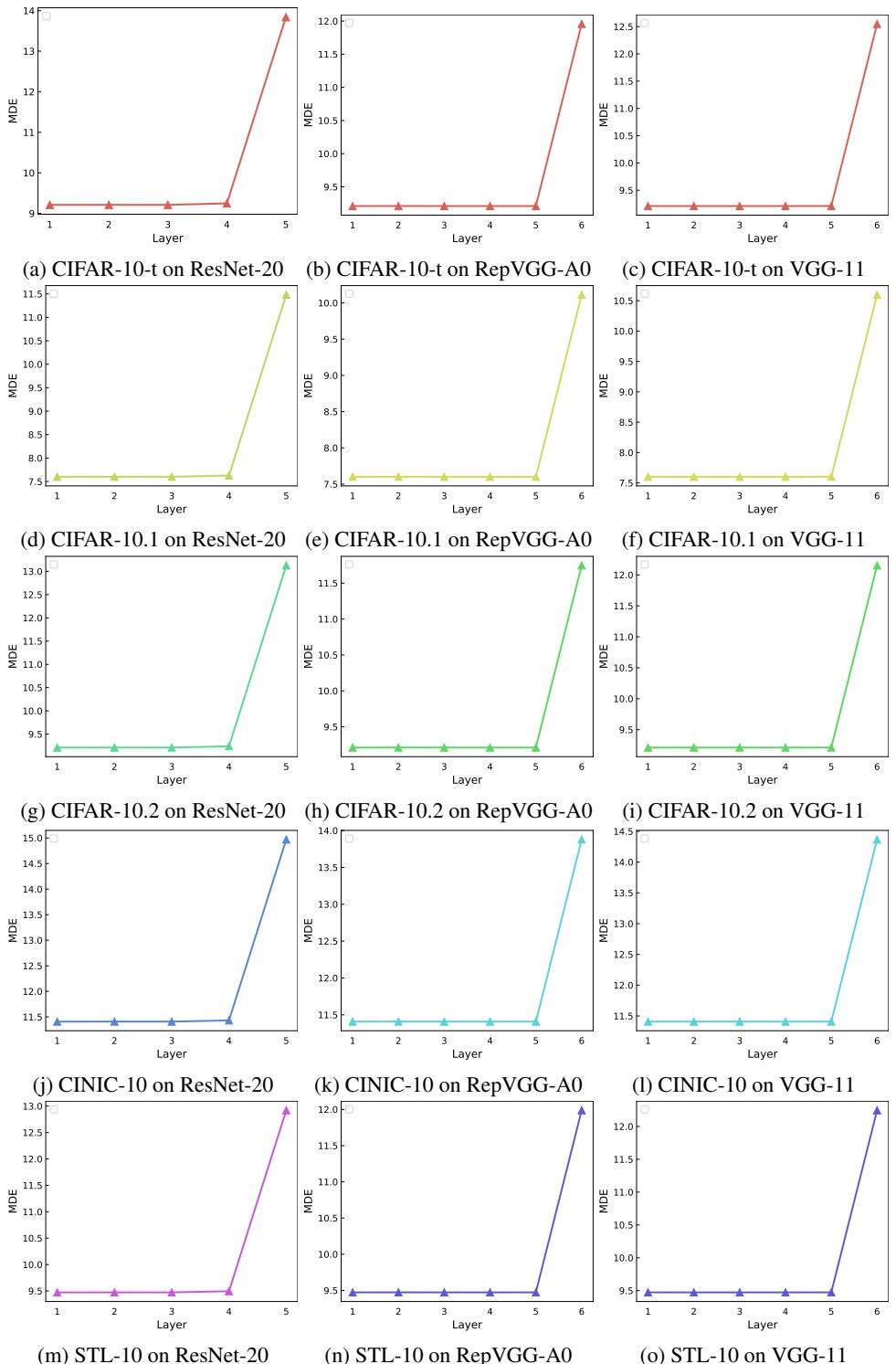

(a) CIFAR-10-t on ResNet-20   (b) CIFAR-10-t on RepVGG-A0   (c) CIFAR-10-t on VGG-11

(d) CIFAR-10.1 on ResNet-20   (e) CIFAR-10.1 on RepVGG-A0   (f) CIFAR-10.1 on VGG-11

(g) CIFAR-10.2 on ResNet-20   (h) CIFAR-10.2 on RepVGG-A0   (i) CIFAR-10.2 on VGG-11

(j) CINIC-10 on ResNet-20   (k) CINIC-10 on RepVGG-A0   (l) CINIC-10 on VGG-11

(m) STL-10 on ResNet-20   (n) STL-10 on RepVGG-A0   (o) STL-10 on VGG-11

Figure 6: MDE scores calculated from the features of different layers in CIFAR-10 Setup.

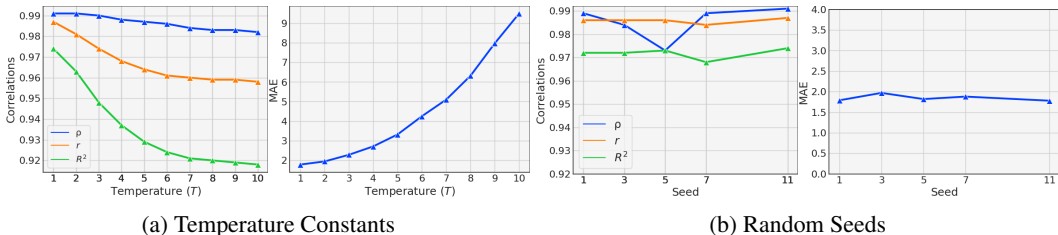

(a) Temperature Constants
(b) Random Seeds

Figure 7: MDE's coefficients of determination ($R^2$), Pearson's correlation ($r$), Spearman's rank correlation ($\rho$) and mean absolute errors (MAE) on *(a)-scaled temperature constants* and *(b)-different random seeds*.

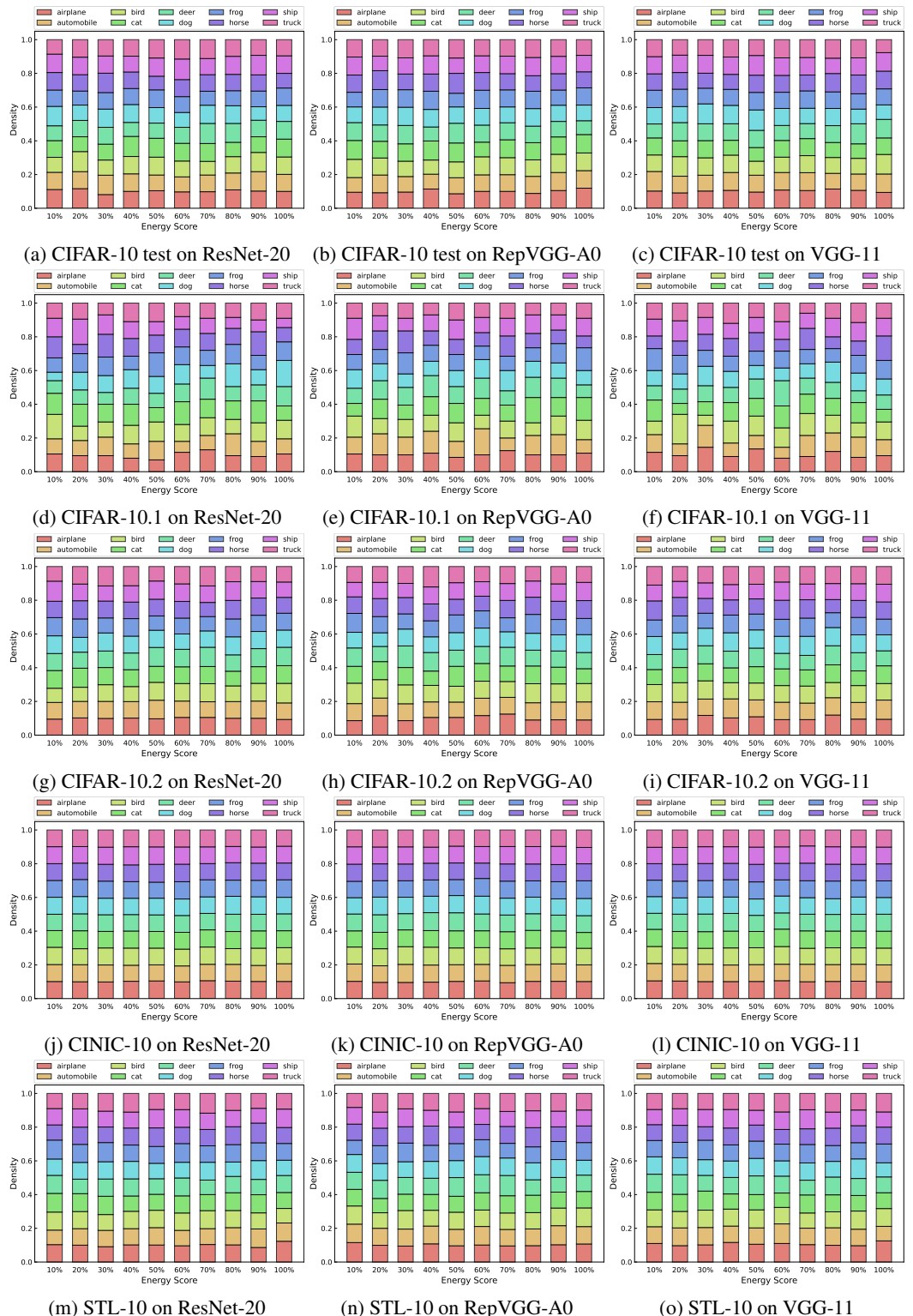

Figure 8: Bucketing of sample categories by energy scores in the CIFAR-10 Setup.

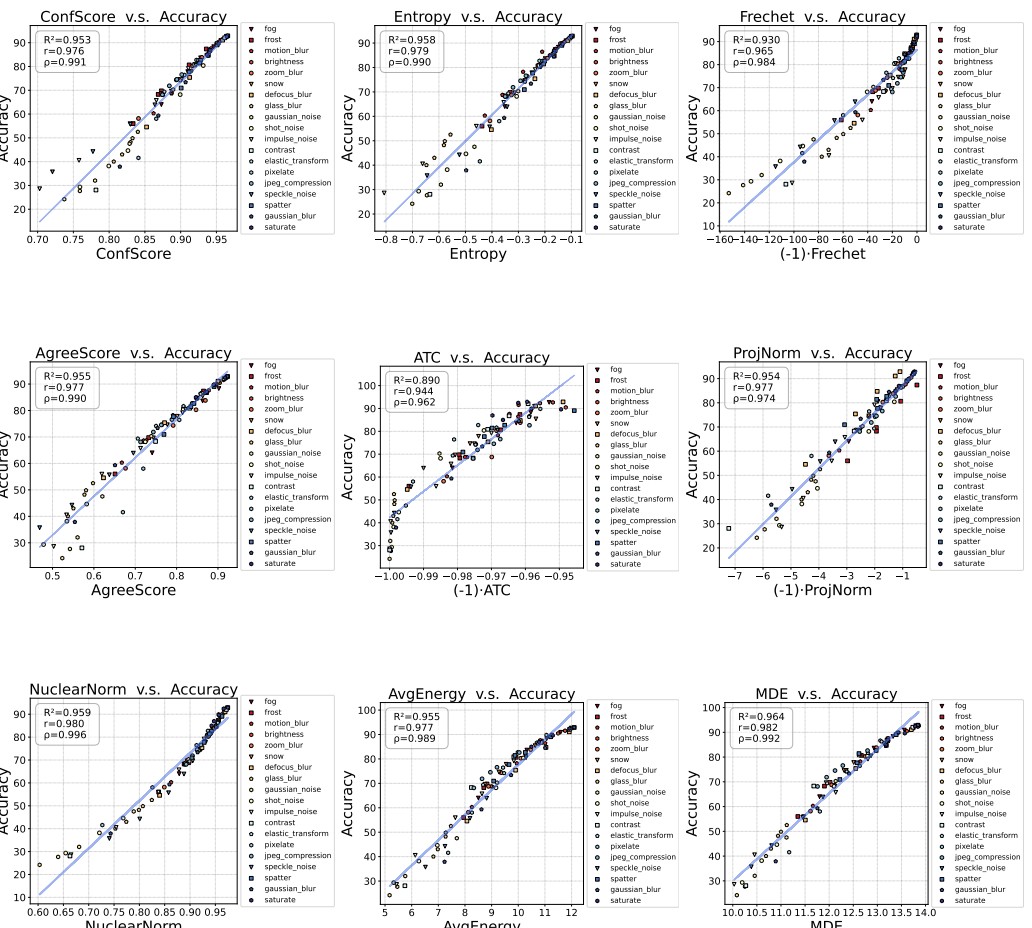

Figure 9: Correlation comparison of all methods using **ResNet-20** on synthetic shifted datasets of **CIFAR-10** setup. We report coefficients of determination ($R^2$), Pearson's correlation ($r$) and Spearman's rank correlation ($\rho$) (higher is better).

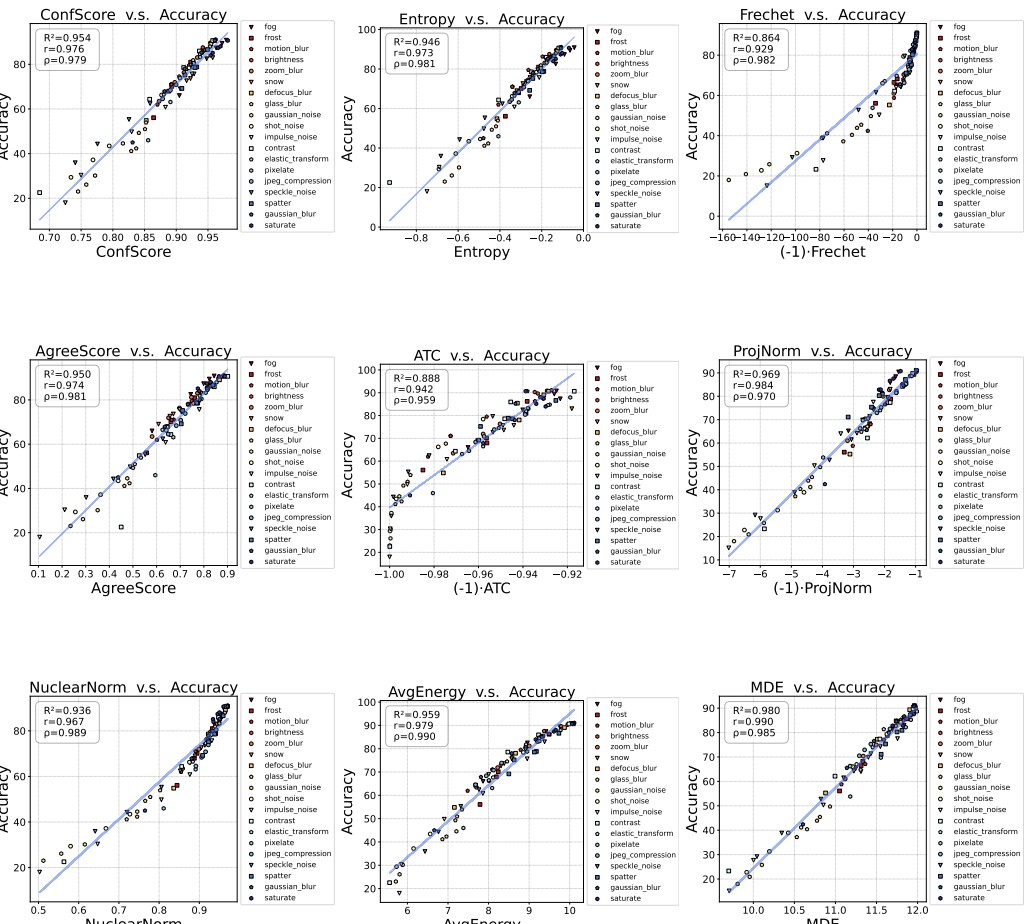

Figure 10: Correlation comparison of all methods using **RepVGG-A0** on synthetic shifted datasets of **CIFAR-10** setup. We report coefficients of determination ($R^2$), Pearson's correlation ($r$) and Spearman's rank correlation ($\rho$) (higher is better).

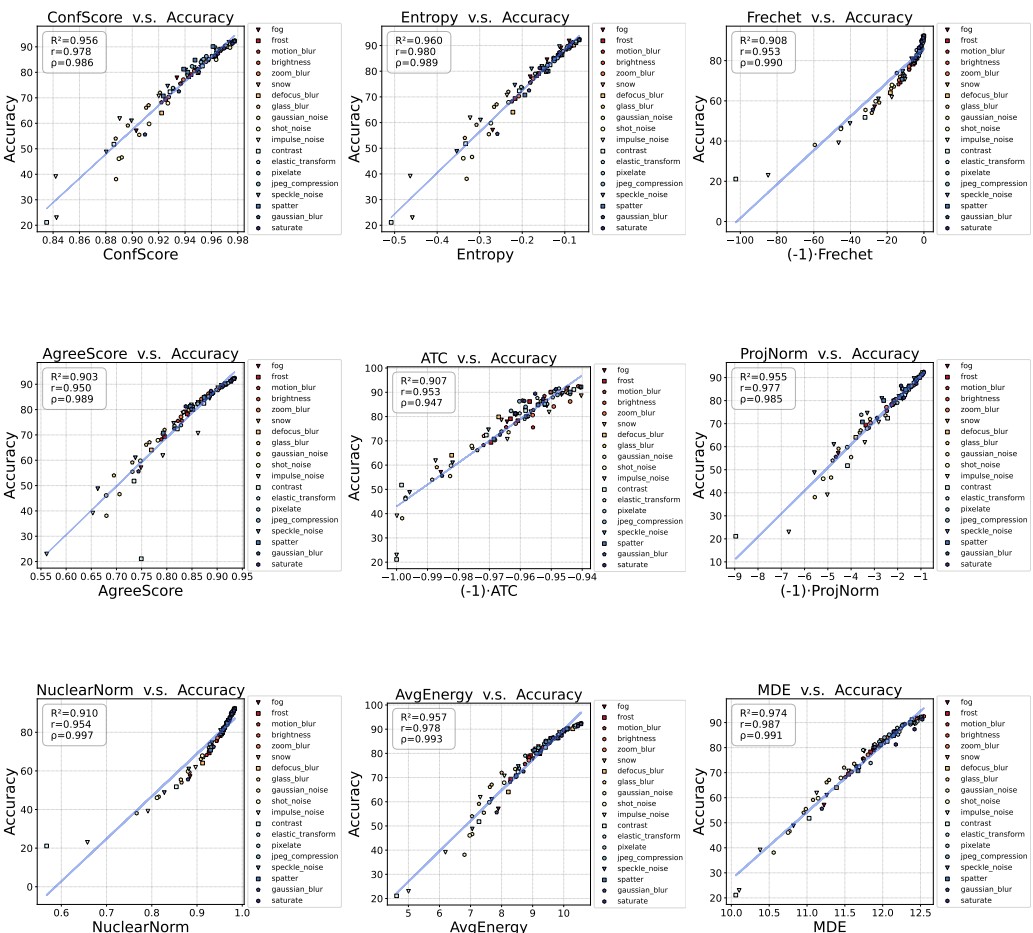

Figure 11: Correlation comparison of all methods using **VGG-11** on synthetic shifted datasets of **CIFAR-10** setup. We report coefficients of determination ($R^2$), Pearson's correlation ($r$) and Spearman's rank correlation ($\rho$) (higher is better).

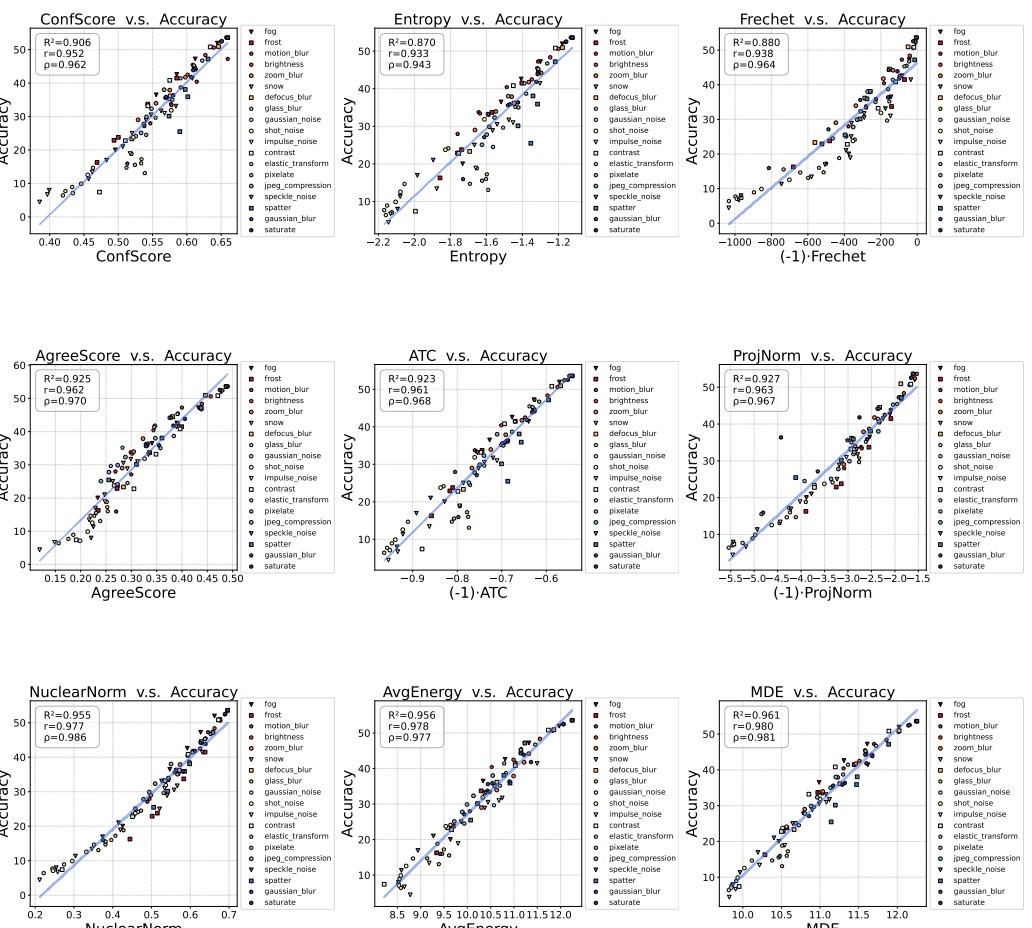

Figure 12: Correlation comparison of all methods using **ResNet-20** on synthetic shifted datasets of **CIFAR-100** setup. We report coefficients of determination ($R^2$), Pearson's correlation ($r$) and Spearman's rank correlation ($\rho$) (higher is better).

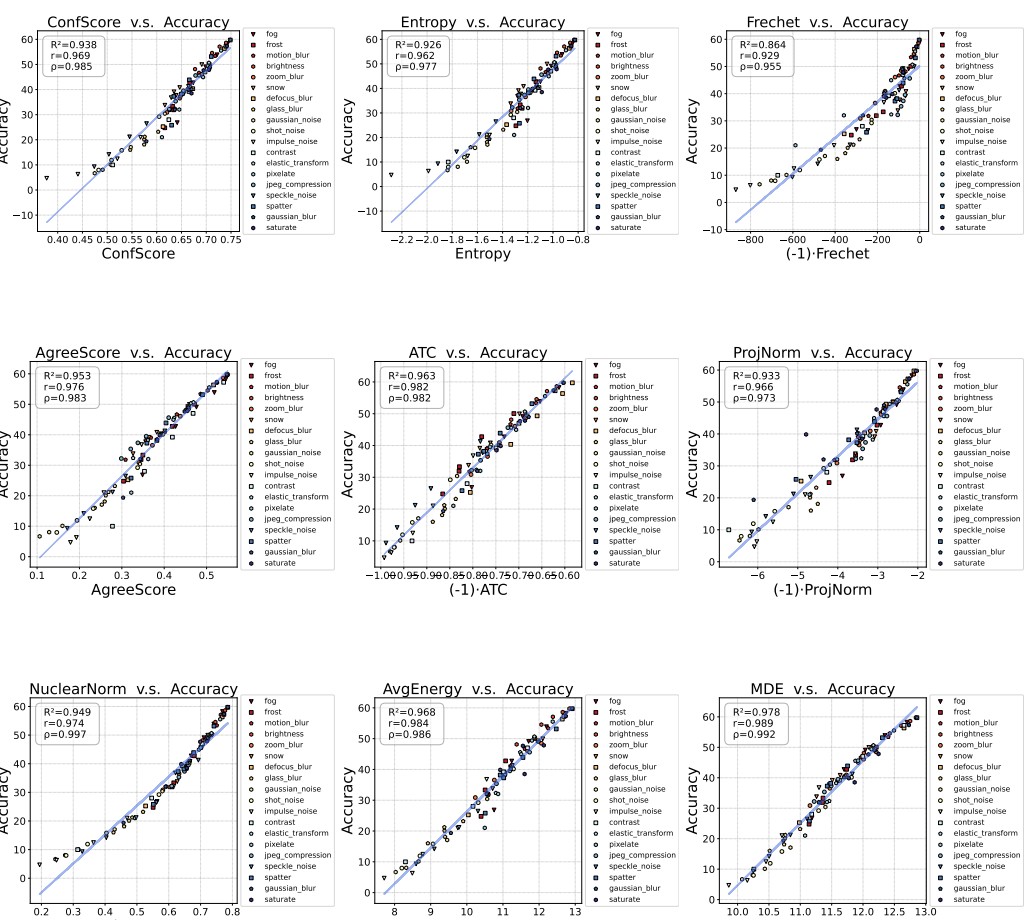

Figure 13: Correlation comparison of all methods using **RepVGG-A0** on synthetic shifted datasets of **CIFAR-100** setup. We report coefficients of determination ($R^2$), Pearson's correlation ($r$) and Spearman's rank correlation ($\rho$) (higher is better).

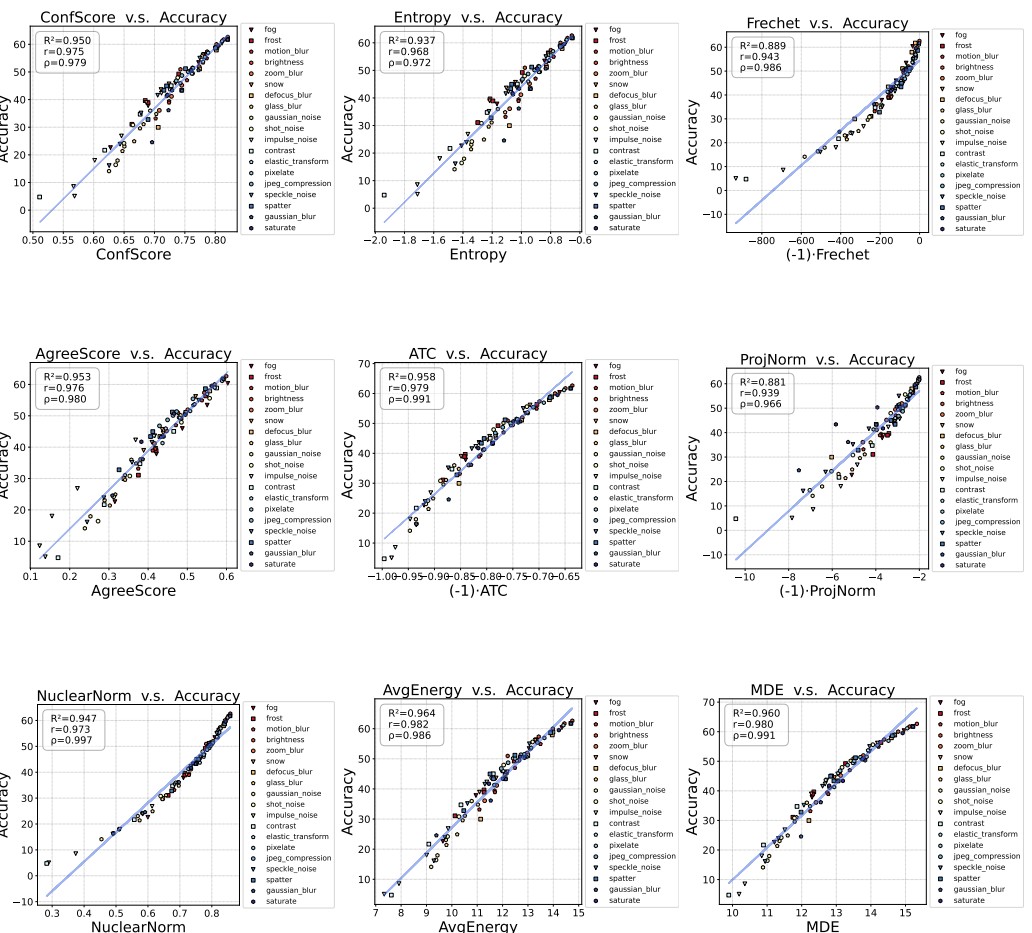

Figure 14: Correlation comparison of all methods using **VGG-11** on synthetic shifted datasets of **CIFAR-100** setup. We report coefficients of determination ($R^2$), Pearson's correlation ($r$) and Spearman's rank correlation ($\rho$) (higher is better).

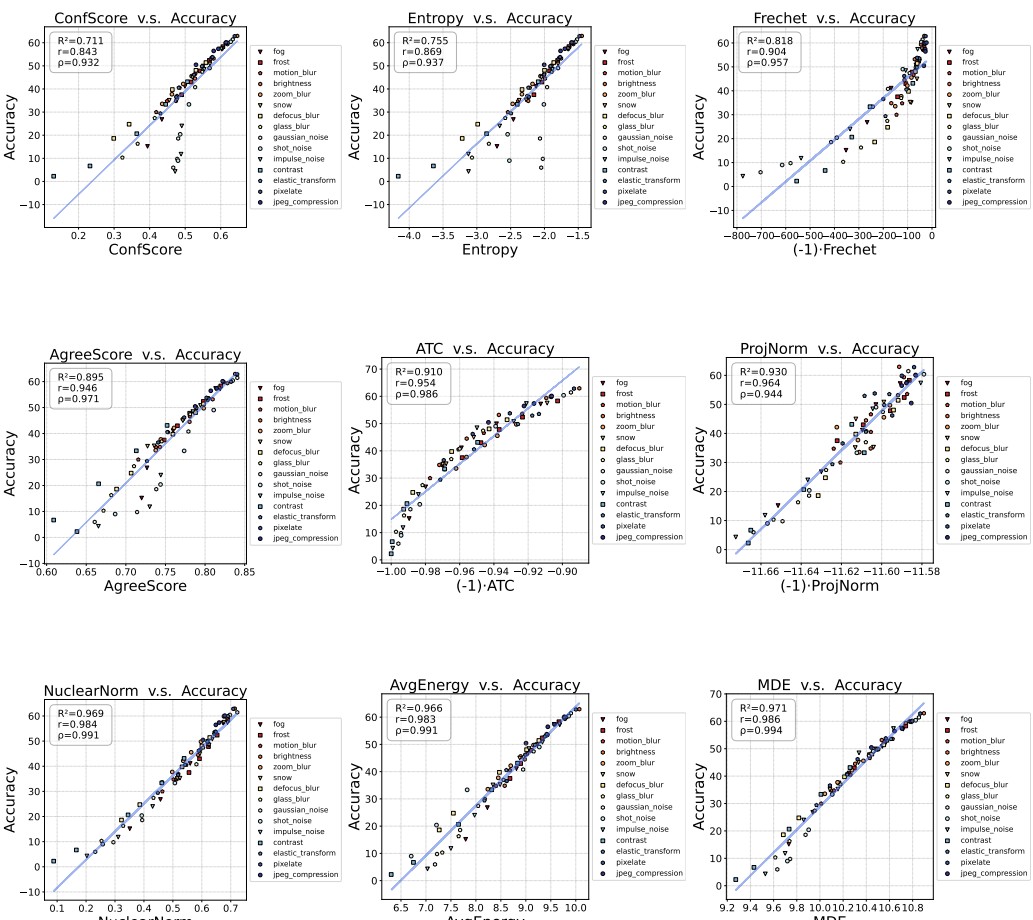

Figure 15: Correlation comparison of all methods using **ResNet-50** on synthetic shifted datasets of **TinyImageNet** setup. We report coefficients of determination ($R^2$), Pearson's correlation ($r$) and Spearman's rank correlation ($\rho$) (higher is better).

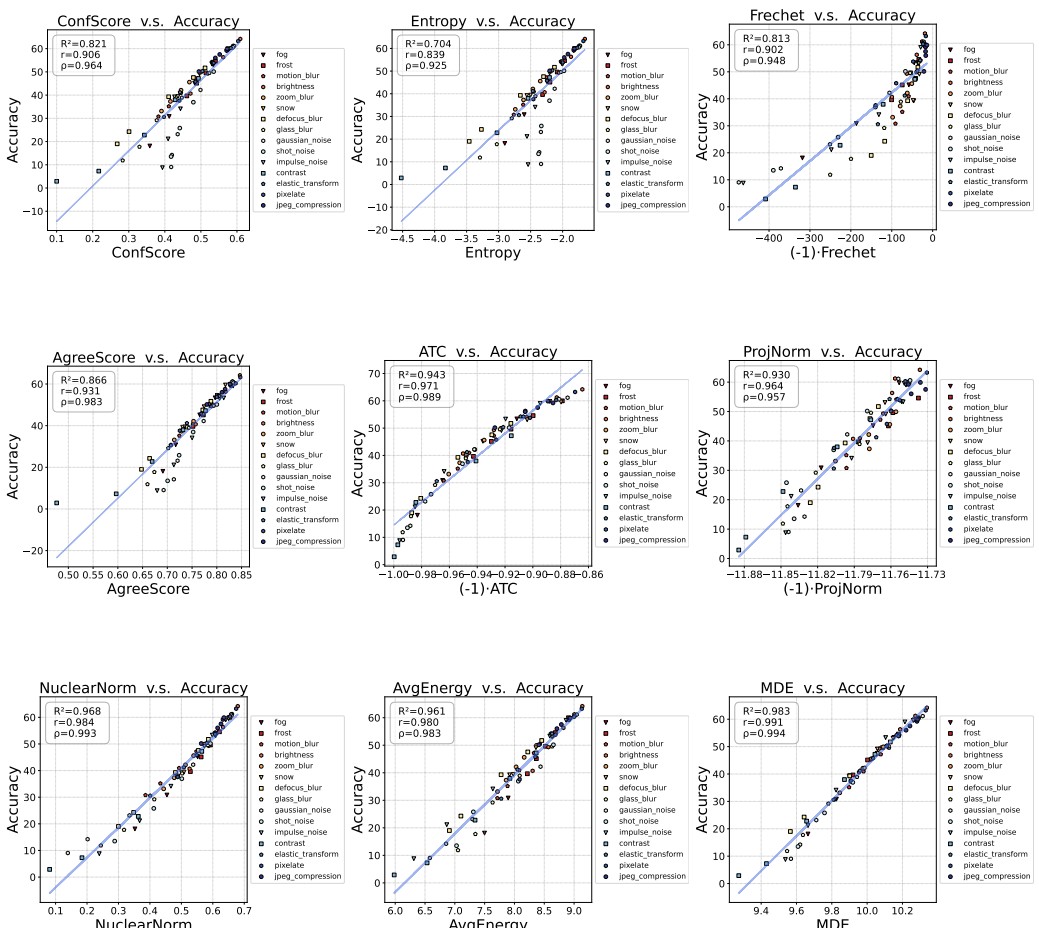

Figure 16: Correlation comparison of all methods using **DenseNet-161** on synthetic shifted datasets of **TinyImageNet** setup. We report coefficients of determination ($R^2$), Pearson's correlation ($r$) and Spearman's rank correlation ($\rho$) (higher is better).

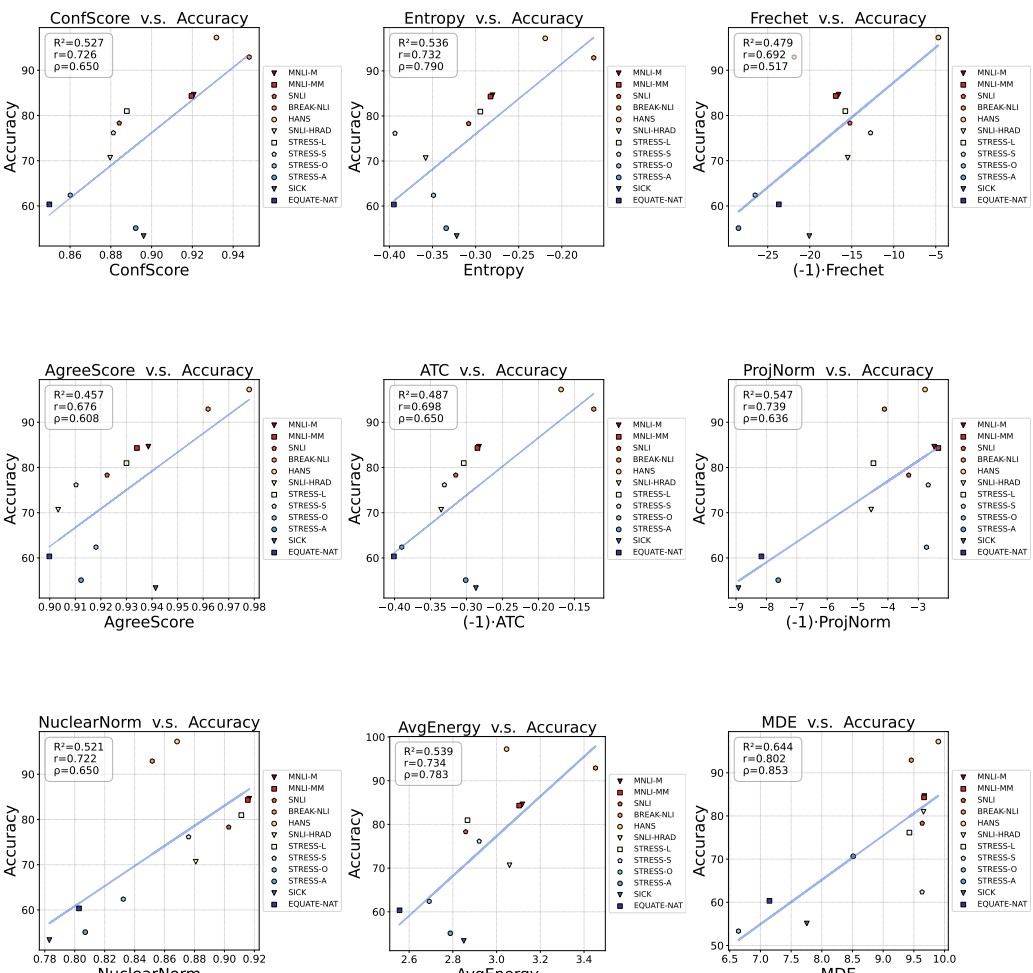

Figure 17: Correlation comparison of all methods using **BERT** on synthetic shifted datasets of **MNLI** setup. We report coefficients of determination ($R^2$), Pearson's correlation ($r$) and Spearman's rank correlation ($\rho$) (higher is better).

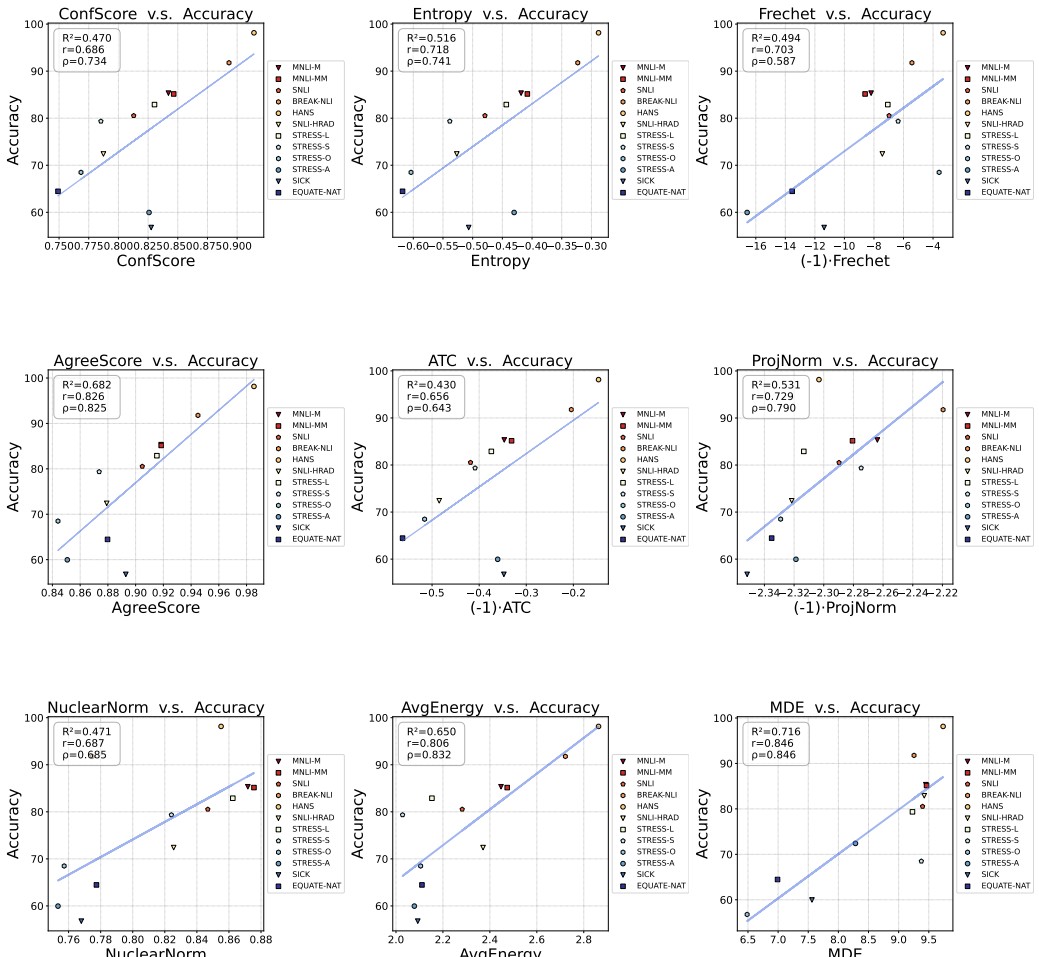

Figure 18: Correlation comparison of all methods using **RoBERTa** on synthetic shifted datasets of **MNLI** setup. We report coefficients of determination ($R^2$), Pearson's correlation ($r$) and Spearman's rank correlation ($\rho$) (higher is better).

