# OpenReview forum: "Energy-based Automated Model Evaluation"
_ICLR.cc/2024/Conference — ICLR 2024 poster_

### Official Review · Reviewer_2CiW · 2023-10-29

**Soundness:** 3 good
**Presentation:** 4 excellent
**Contribution:** 3 good
**Rating:** 8
**Confidence:** 3

**Summary:**

This paper introduces a metric called Meta-Distribution Energy (MDE) for automated model evaluation (assessing model performance on unlabeled datasets). The method involves extracting energy scores from the model's output and extending them into a statistical representation at the dataset level. The paper demonstrates a strong linear relationship between MDE and classification accuracy on datasets with differing extents of distribution shifts. This allows for predicting model performance based on the MDE of an unlabeled test set, and the authors provide theoretical proof of this concept. In the experimental section, the paper presents the outstanding performance of MDE across different backbones, datasets, and modalities, even in scenarios with noise and class imbalance.

**Strengths:**

1) The task of AutoEval is crucial when deploying neural network models in real-world scenarios. It can help avoid the issue of not knowing the actual performance in practical applications and is also beneficial for model selection."
2) The method MDE in the paper exhibits a relatively high level of novelty by introducing the statistical distribution of energy scores into the field of AutoEval for the first time. Furthermore, the paper extends the approach to improving energy scores, which can be applied to a wider range of tasks. Additionally, the authors provide detailed theoretical proof that strongly illustrates the effectiveness of the method.
3) The paper conducts extensive experiments on different backbones, different datasets, and different modalities. The experimental results indicate a significant performance improvement compared to other methods in the field.
4) The paper is of high writing quality, with a smooth and easy-to-grasp presentation of the key points.

**Weaknesses:**

1) It seems that the paper does not provide a more detailed explanation of the parameter T in MDE. Is T just an ordinary parameter, or does it have a more intuitive meaning? In Figure 3(a), only the trend of T from 1 to 10 is displayed. What is the impact of a wider range of T values on the results?
2) In the paper, the authors categorized relevant methods into 'training-free' and 'training-must.' How much progress does MDE, as a 'training-free' method, make compared to 'training-must' methods in terms of evaluation time and memory usage? It would be ideal to conduct an experiment to illustrate this.

**Questions:**

Please try to address the questions I raised in the 'Weaknesses' chapter.

---

> ### Author Response · Authors · 2023-11-20
> **Author Response to Reviewer 2CiW**
>
> We appreciate your positive comments and valuable suggestions.
>
> **W1: Concise description of temperature and small range temperature ablation**
>
> Thanks for your advice! The parameter $T$ enables control over the smoothness of the model's probability distribution during generation.
>
> The ablation results for a wide range of temperature hyperparameter (**$T$ from 0.1 to 100**) are shown below:
>
> | T | 0.01 | 0.5 | 1 | 2 | 3 | 4 | 5 | 6 | 7 | 8 | 9 | 10 | 20 | 30 | 40 | 50 | 60 | 70 | 80 | 90 | 100 |
> | :---: | :---: | :---: | :---: | :---: | :---: | :---: | :---: | :---: | :---: | :---: | :---: | :---: | :---: | :---: | :---: | :---: | :---: | :---: | :---: | :---: | :---: |
> |  $ρ$ | 0.988 | 0.989 | 0.991 | 0.991 | 0.990 | 0.988 | 0.987 | 0.986 | 0.984 | 0.983 | 0.983 | 0.982 | 0.976 | 0.974 | 0.974 | 0.974 | 0.973 | 0.971 | 0.972 | 0.971 | 0.971 |
> | $r$ | 0.989 | 0.988 | 0.987 | 0.981 | 0.974 | 0.968 | 0.964 | 0.961 | 0.960 | 0.959 | 0.959 | 0.958 | 0.959 | 0.959 | 0.959 | 0.960 | 0.960 | 0.960 | 0.960 | 0.960 | 0.961 |
> | $r^2$ | 0.978 | 0.977 | 0.974 | 0.963 | 0.948 | 0.937 | 0.929 | 0.924 | 0.921 | 0.920 | 0.919 | 0.918 | 0.919 | 0.920 | 0.921 | 0.921 | 0.921 | 0.921 | 0.922 | 0.922 | 0.923 |
> | MAE | 1.72 | 1.80 | 1.78 | 1.94 | 2.28 | 2.71 | 3.32 | 4.25 | 5.10 | 6.32 | 7.98 | 9.50 | 9.87 | 10.05 | 10.16 | 10.22 | 10.24 | 10.25 | 10.25 | 10.24 | 10.26 |
>
> In fact, despite the ideal case for MDE being T→0 in Theorem 3.1, our empirical results found that the performance variation caused by T from 0 to 1 is minimal.
> Thus, we fix T = 1 to ensure that the classification accuracy is not affected so that we can compare fairly with baselines. Most importantly, we did **not rely on T to tune performance**.
>
> **W2: Lacking a comparison in terms of evaluation time and memory usage between MDE and the training-must method**
>
> Honestly speaking, it is inaccessible to fairly compare the running time and memory usage of different methods.
>
> Because **the workflow of different methods are very different**, e.g. training model from scratch, fine-tuning model, calculating model feature, statistic dataset distribution, computing the disagreement of ensemble prediction, etc.
>
> Intuitively, it is easy to understand because the training-must method requires retraining the model, which can consume huge memory and require much more time, while MDE only requires computing values from the model logits.
>
> So, we **simplify this problem** to **compare the time complexity and space complexity** of different methods in a rough granularity:
>
> 1. **For time complexity**:
>
>     *AC = ANE < ATC < AvgEnergy = MDE(ours) = NuclearNorm < Frechet < ProjNorm(training-must) < AgreeScore(training-must)*
>
> 2. **For space complexity**:
>
>     *training-free(AC, ANE, ATC, AvgEnergy, MDE(ours), NuclearNorm, Frechet) < training-must(ProjNorm, AgreeScore)*
>
> We hope our response addresses your original questions. Please let us know if you have any other comments. We are eager to hear your feedback.

---

### Official Review · Reviewer_Dsfd · 2023-10-29

**Soundness:** 4 excellent
**Presentation:** 3 good
**Contribution:** 3 good
**Rating:** 8
**Confidence:** 4

**Summary:**

This paper proposes a novel measure called Meta-Distribution Energy (MDE) for automated model evaluation (AutoEval), which is a method for assessing the performance of machine learning models on unlabeled test data. The core idea of MDE is to convert the energy information of the model's output into a probability distribution statistic, which enables a smoother data representation. The paper also provides theoretical analysis connecting MDE with classification loss, proving its effectiveness. Experimental results demonstrate MDE's superior performance across various modalities, datasets, and model architectures, especially in noisy and class-imbalanced scenarios.

**Strengths:**

* The research topic of this paper -- AutoEval is valuable in the real world deployment of DNNs. I believe this area may shed some novel light on the unsupervised evaluation community.
* This paper establishes a connection between MDE and classification loss through a mathematical theorem, providing theoretical justification for the effectiveness of the proposed method.
* MDE demonstrates strong performance across a variety of modalities, datasets, and model architectures, and these detailed analyzes make it as a versatile solution for automated model evaluation.
* MDE remains effective even in challenging scenarios such as strong noise and class imbalance, showcasing its robustness in practical applications.

**Weaknesses:**

* The paper does not provide source code, making re-implementation challenging. Please provide the code later to dispel this concern.
* Interpretability can be further explored. While MDE provides strong correlation with classification accuracy, further research could focus on enhancing the interpretability of the method, making it easier for users to understand and trust the results.
* Different methods seems to have different (wall-clock?) time required to come up with such evaluation -- perhaps some notes on that would be helpful as well.
* Grammar should be thoroughly checked. For example, in the first paragraph, "the information (energy) associated with individual samples, then offer a smoother representation enabled by energy-based learning." should be changed to "xxxx, and then offers...". Similarly, in the second paragraph, "the correct classified data are given low energies, and vice versa" should be changed to "the correctly classified..."

**Questions:**

Please refer to the weakness section mentioned above.

---

> ### Author Response · Authors · 2023-11-20
> **Author Response to Reviewer Dsfd**
>
> We express our gratitude for favorable feedback and valuable insights, and we sincerely hope that our response can address your concern.
>
> **W1: No source code provided.**
>
> We apologize for the misunderstanding caused by not mentioning in the paper that we have provided our source code in the supplementary materials.
>
> **W2: Interpretability can be further explored.**
>
> Thanks for your comments. We will engage in further discussions to enhance the interpretability of our methods.
>
> For instance, we will introduce more easily comprehensible theoretical proofs, provide concrete examples/illustrations, and showcase a greater number of visual experiments in our future work.
>
> **W3: The comparison of time costs among different methods is essential.**
>
> Honestly speaking, it is impossible to fairly compare wall clocks of different methods.
>
> Because **the workflow of different methods is very different**, e.g. training the model from scratch, fine-tuning the model, calculating model features, statistic dataset distribution, computing the disagreement of ensemble prediction, etc.
>
> So we **simplify this problem** to **compare the time complexity** of different methods in a rough granularity:
>
> 1. From the view of method categorization:
>
>     *confidence-based < regression-based (we are here) < based on data distribution shift = based on model feature < based on model parameter shift < disagreement-based = based on self-supervised task*
>
> 2. From the view of the compared baseline in this article:
>
>     *AC = ANE < ATC < AvgEnergy = MDE (ours) = NuclearNorm < Frechet < ProjNorm < AgreeScore*
>
>
> **W4: Grammar should be thoroughly checked.**
>
> We will thoroughly check the grammar of the entire article based on your suggestions.

---

### Official Review · Reviewer_SNMZ · 2023-11-02

**Soundness:** 2 fair
**Presentation:** 2 fair
**Contribution:** 1 poor
**Rating:** 3
**Confidence:** 5

**Summary:**

This paper focuses on AutoEval, where the goal is to estimate classifier performance on unlabeled test sets. To this end, this work proposes to use energy score (MDE) as the statistic of each test set for inferring the corresponding classification accuracy. The experimental results on several benchmarks such as CIFAR-10 and CIFAR-100 show the proposed can achieve reasonable accuracy estimation.

**Strengths:**

+ Algorithm 1 clearly shows the proposed method. The whole pipeline is well-introduced
+ The experiment includes MNLI, a natural language inference task.

**Weaknesses:**

- *1. Overstated claims*: The paper asserts that "the AutoEval frameworks still suffer from an overconfidence issue" without providing clear, empirical examples. Additionally, the statement regarding "substantial storage" lacks a comparison with existing methods such as DoC and ATC, leaving the reader unconvinced of any real advantage of the proposed method. Furthermore, the claim that the proposed MDE method is superior in terms of "computational cost" is not substantiated, especially considering that unlike ATC, MDE requires the training of a linear regression model. To move forward, it would be essential to clarify these issues with specific data and comparative analysis.

- *2. Limited contribution*: The novelty of your work is questioned due to the similarity with existing methods in the field. The use of Energy score as a replacement for Softmax score, without significant insights or enhancements, is seen as insufficient for constituting a substantial contribution. Prior works [Detecting Errors and Estimating Accuracy on Unlabeled Data with Self-training Ensembles] have already discussed the connection between OOD detection and AutoEval, which the current submission does not appear to extend beyond.

- *3. Unconvincing theoretical analysis*: The theoretical grounding provided in Section 3.3 is deemed unclear and unconvincing. The methodology for ascertaining model accuracy needs to account for scenarios where an image is correctly classified with a low Softmax score. A more robust theoretical framework is necessary to support the claims made.

- *4. Experimental results are not solid*: The absence of results for ImageNet-1K in Table 1, along with other test datasets like ImageNet-S/A/V2, raises concerns about the comprehensiveness of the experimental evaluation. Moreover, the omission of recent relevant works, such as "Characterizing Out-of-Distribution Error via Optimal Transport," and the lack of comparison with methods like Nuclear Norm on ImageNet setup, call into question the thoroughness of the analysis.

- *5. Limited Dataset Diversity*: The paper does not report on more natural shifts, which are included in benchmarks provided by methods like ATC. While BREEDs are mentioned in the supplementary material, results are not presented, and expectations for insights on datasets like i-WILDS are not met.

**Questions:**

- Please clarify the claims on overconfidence issue, substantial storage, and computation cost
- Please report the results on the ImageNet setup and the estimates of ImageNet-A/V2/S
- Please clearly compare with existing works such as ATC, DoC, and Nuclear Norm on ImageNet. Moreover, [Characterizing Out-of-Distribution Error via Optimal Transport] should be included for comparison.
- Other related works: Estimating and explaining model performance when both covariates and labels shift. In NeurIPS 2022

---

> ### Author Response · Authors · 2023-11-20
> **Author Response to Reviewer SNMZ (Part I)**
>
> We would like to thank you for your constructive comments, which are very helpful and make our paper even stronger! We sincerely hope that our response addresses your concerns.
>
> **W1&Q1: Please clarify the claims on the overconfidence issue, substantial storage, and computation cost**
>
> We would like to give a clearer clarification of the advantages of MDE compared to existing AutoEval methods from three aspects:
>
> **i. Overconfidence issue:**
>
> For methods based on model confidence, such as ConfScore, Entropy, DoC [1], and ATC, we provide an intuitive example to illustrate why they suffer from overconfidence issues.
>
> Given two samples with labels [0, 1, 0] and [0, 1, 0], a model outputs softmax scores for the two samples such as [0.8, 0.1, 0.1] and [0.1, 0.8, 0.1].
> At this point, the model's confidence is 80%, while the prediction accuracy is 50%, and the MAE between the two quantities is 30%. The model overestimates the prediction truthiness of its softmax probability, i.e., "overconfidence”.
>
> Therefore, it is necessary to introduce model calibration techniques. We adopt commonly used temperature scaling (i.e. dividing logits by the coefficient $t$ before applying Softmax) in the ATC paper for demonstration:
>
> Let $t$ =2, the two softmax scores will be transformed to [0.4, 0.05, 0.05] and [0.05, 0.4, 0.05]. Now the model's Softmax confidence is 40%, while the prediction accuracy remains at 50%, the MAE will decrease by 20% for accuracy prediction.
>
> However, MDE is a meta-distribution statistical metric based on the sample’s energy (a non-normalized scalar), which does not depend on Softmax confidence and thus is not affected by overconfidence.
>
> **ii. Substantial storage and Computation cost:**
>
> 1. For methods based on prediction disagreement, such as self-training ensembles [2]; and AgreeScore, they generally use the disagreement of the prediction outputs of two or even multiple models for accuracy predictions. They **additionally train or fine-tune models and store these models**, which incurs additional storage and computing overhead.
> 2. For methods based on model parameters, such as ProjNorm, first use the model to label the target data, then fine-tune the model on the pseudo data and finally calculate the model parameter shift to predict accuracy, while Weights [3] and Norm-based metrics [4] estimate the accuracy through the norm of the model weight matrix and bias vector. These methods **either fine-tune the model or use model parameters to calculate**, which is quite a bit of storage burden and computational cost.
> 3. For methods based on self-supervised tasks, such as Rotation [5] and CAME [6], they jointly train the model from scratch by self-supervised task together with the classification task, and use the self-supervised accuracy as a proxy for classification accuracy. This way of **retraining the model** inevitably introduces extra computational overhead.
> 4. For methods based on dataset distribution shift, such as Frechet and Bag-of-prototypes [7], when predicting the model accuracy of the target dataset, they need to **access the training set, obtain model features, and calculate the mean of feature or cluster the features**, this sort of method also consumes computing resources in practice.
>
> However, MDE only requires the model logits on the target dataset for calculation, without the above highlighted constraints. This effectively reduces the substantial storage and computation cost.
>
> **iii. Compared with ATC, MDE requires training a linear regression model**
>
> 1. The buildup of **fitting regression models** via strong correlations between unsupervised indicator and accuracy, which **is just a common workflow** for several published AutoEval works ([4], [5], ProjNorm, Nuclear Norm, Frechet, Bag-of-prototypes, etc), no exclusive to MDE. In practice, we only call the `sklearn.linear_model.LinearRegression`to fit the regression model, which is almost not time-consuming.
> 2. The ATC method requires identifying a threshold on the source validation data and performing model calibration to obtain better accuracy predictions, it also has its own computational overhead.
> 3. Most importantly, **the performance of MDE significantly surpasses ATC**. Compared with ATC, the mean absolute error of MDE dropped a lot (CIFAR-10: 8.15→1.78, TinyImageNet: 11.37→2.15, ImageNet: 11.51→5.48, MNLI: 15.77->5.76, Wilds: 6.35→2.26), the correlation also improves especially for MNLI ($\rho$: 0.647→0.850, $R^2$: 0.459→0.680).

---

> ### Author Response · Authors · 2023-11-20
> **Author Response to Reviewer SNMZ (Part II)**
>
> **W2: Please clearly explain the contributions of this paper.**
>
> We respectfully disagree with the comment that our work has limited contributions, the following statements are intended to dispel the concerns raised by the reviewers:
>
> **i. The use of Energy score as a replacement for Softmax score：**
>
> 1. **_MDE and Softmax score are different in essence and usage._**
>
> - *Essence*: MDE is defined as the meta-distribution statistic of non-normalized energy at the dataset level, while Softmax score is the maximum value of the normalized logit vector for a single sample.
>
> - *Usage*: Softmax score typically reflects the classification accuracy through measures such as the mean, mean difference, or the data proportion below a certain threshold. On the other hand, MDE predicts accuracy by a regression model.
>
> 2. **_MDE and Softmax score have different mathematical forms._**
> \begin{equation}
> \mathcal{MDE}(x;f) =-\frac{1}{|N|} \sum_{i=1}^N \log \operatorname{Softmax} E(x;f) =-\frac{1}{|N|} \sum_{i=1}^N \log \frac{e^{E\left(x_n;f\right)}}{\sum_{i=1}^N e^{E\left(x_i;f\right)}},
> \end{equation}
> \begin{equation}
> \max _y p(y \mid \mathbf{x})=\max _y \frac{e^{f_y(\mathbf{x})}}{\sum_i e^{f_i(\mathbf{x})}}=\frac{e^{f^{\max }(\mathbf{x})}}{\sum_i e^{f_i(\mathbf{x})}}
> \end{equation}
>
> 3. **_MDE is more suitable for accuracy prediction than Softmax score_**
>
>     To demonstrate this, we decompose the softmax confidence by logarithmizing it as follows:
>     \begin{equation}
>     \begin{aligned}
>     \log \max _y p(y \mid \mathbf{x}) & =E\left(\mathbf{x} ; f(\mathbf{x})-f^{\max }(\mathbf{x})\right) \\\\
>     & =E(\mathbf{x} ; f)+f^{\max }(\mathbf{x})
>     \end{aligned}
>     \end{equation}
>
>      Then, we find $f^{max}(x)$ tends to be lower and $E(x; f)$ tends to be higher for OOD data, and vice versa. This shifting results in Softmax confidence that is no longer suitable for accuracy prediction, while MDE is not affected by this bothersome issue.
>
> **ii. Existing work has discussed the connection between OOD detection and AutoEval, and this paper does not extend this scope.**
>
> 1. This paper **does not explore the relationship** between OOD detection and AutoEval but rather proposes **a novel AutoEval method**.
>
> 2. AutoEval has fundamental differences with OOD detection. OOD detection is to **identify outlier test samples** that are different from training distributions, but AutoEval is towards **unsupervised estimating of the model’s testing accuracy.**
>
> **iii. Reviewers *Dsfd* and *2CiW*** both **agree** that the MDE method has **theoretical interpretability and practical application effectiveness**. **Reviewer 2CiW** also **acknowledged the relatively high novelty** of the MDE method.
>
> **W3: Unconvincing theoretical analysis**
>
> We respectfully disagree with this comment.
> Considering **the scenario** mentioned by the reviewer where **the model accurately classifies images with lower softmax scores**, we provide the following example to illustrate how the Eq.(9) of Theorem 3.1 substantiates the MDE can work in such a case.
> \begin{align}
> \Delta^{i}=\mathcal{MDE}^i - \mathcal{L}^i_{\text {nll }}
> = f_{y_i}({x_i})/T - \max _{j \in \mathcal{Y}}{f_j\left(x_i\right)/T} =\begin{cases}0, & \text { if } j=y_i, \\\\
> <0, & \text { if } j \neq y_i,\end{cases},   \quad\quad(9)
> \end{align}
>
> Given a sample with the label [0, 1, 0],  the model outputs the Softmax score for this sample as [0.4, 0.5, 0.1].
> According to Theorem 3.1, we know that $y_i$ =2, $j$ corresponding to $\max _{j \in \mathcal{Y}}{f_j\left(x_i\right)/T}$  is 2, so the model classifies correctly with a lower softmax score.
>
> In turn, **when the model incorrectly classifies images with higher softmax scores**, that is given a sample with the label [0, 1, 0], the model outputs the Softmax score for this sample as [0.5, 0.4, 0.1].
> According to Theorem 3.1, $y_i$ is still 2, but $j$ becomes 1, so $j$ is not equal to $y_i$, and $f_{y_i}\left(x_i\right) / T-\max _{j \in \mathcal{Y}} f_j\left(x_i\right) / T = 0.4 - 0.5 $ < 0.
>
> From the above examples, by judging whether the term of Eq. 9 is less than 0, we can know whether the index $j$ belongs to the maximum logits equal to the label $y$, and then we can obtain the accuracy of the
> classifier $f$. Thus, we mathematically establish a connection between MDE and classification risk.
>
> Moreover, both **Reviewers *Dsfd* and *2CiW* concurred** that **the theoretical proof** of this paper **has substantiated the effectiveness** of the proposed method.

---

> ### Author Response · Authors · 2023-11-20
> **Author Response to Reviewer SNMZ (Part III)**
>
> **W4&Q2&Q3: Please report the results on the ImageNet-1k setup and compare MDE with existing works**
>
> Following the reviewer’s advice, we add two tables that compare the correlation and MAE with existing methods **(including COT [8])** on the ImageNet-1K setup.
>
> **Table 1. Correlation comparison with existing methods on ImageNet Setup**
>
> | Dataset | Network | ConfScore |  | Entropy |  | Frechet |  | ATC |  | COT |  | NuclearNorm |  | AvgEnergy |  | MDE |  |
> | :---: | :---: | :---: | :---: | :---: | :---: | :---: | :---: | :---: | :---: | :---: | :---: | :---: | :---: | :---: | :---: | :---: | :---: |
> |  |  |  $ρ$ | $r^2$ |  $ρ$ | $r^2$ |  $ρ$ | $r^2$ |  $ρ$ | $r^2$ |  $ρ$ | $r^2$ |  $ρ$ | $r^2$ |  $ρ$ | $r^2$ |  $ρ$ | $r^2$ |
> | ImageNet | ResNet-152 | 0.980 | 0.949 | 0.979 | 0.946 | 0.945 | 0.879 | 0.980 | 0.899 | 0.968 | 0.943 | 0.979 | 0.961 | 0.980 | 0.955 | **0.982** | **0.967** |
> |  | DenseNet-169 | 0.983 | 0.953 | 0.981 | 0.931 | 0.942 | 0.878 | 0.982 | 0.891 | 0.963 | 0.934 | 0.981 | 0.956 | 0.984 | 0.955 | **0.985** | **0.960** |
> |  | VGG-19 | 0.966 | 0.933 | 0.968 | 0.909 | 0.978 | 0.910 | 0.975 | 0.886 | 0.981 | 0.926 | 0.978 | 0.949 | 0.980 | 0.950 | **0.987** | **0.952** |
> |  | Average | 0.976 | 0.945 | 0.976 | 0.929 | 0.955 | 0.889 | 0.979 | 0.892 | 0.971 | 0.934 | 0.979 | 0.955 | 0.981 | 0.953 | **0.985** | **0.960** |
>
> **Table 2. Mean Absolute Error (MAE) comparison with existing methods on ImageNet Setup**
>
> | Dataset | Unseen test set | ConfScore | Entropy | Frechet | ATC | COT | NuclearNorm | AvgEnergy | MDE |
> | :---: | :---: | :---: | :---: | :---: | :---: | :---: | :---: | :---: | :---: |
> | ImageNet | ImageNet-V2-A | 8.76 | 9.72 | 6.36 | 6.88 | 7.04 | 4.04 | 3.58 | **2.21** |
> |  | ImageNet-V2-B | 8.59 | 10.20 | 9.51 | 8.82 | 9.02 | 5.20 | 4.41 | **2.35** |
> |  | ImageNet-V2-C | 14.92 | 10.27 | 9.12 | 8.74 | 8.85 | 5.66 | 5.01 | **4.27** |
> |  | ImageNet-S | 7.30 | 9.50 | 10.45 | 9.81 | 8.11 | 7.77 | 6.40 | **5.31** |
> |  | ImageNet-R | 15.41 | 13.63 | 12.06 | 12.70 | 12.87 | 11.34 | 10.64 | **7.68** |
> |  | ImageNet-Vid | 15.18 | 14.26 | 16.53 | 14.42 | 13.49 | 12.77 | 11.19 | **8.09** |
> |  | ImageNet-Adv | 19.09 | 18.28 | 18.37 | 19.20 | 14.51 | 13.61 | 11.43 | **8.42** |
> |  | Average | 12.75 | 12.27 | 11.77 | 11.51 | 10.56 | 8.63 | 7.52 | **5.48** |
>
> In summary, MDE demonstrates significant performance gains over existing AutoEval methods (including COT) on ImageNet-1K setup, further confirming the effectiveness of MDE.
>
> **Some remarks:**
>
>   1.  The COT method **requires pre-calibration** of the classifier, which is not a constraint in MDE.
>
> 1. The COT paper was **accepted** in Neurips 2023 **on October 27,** while the **submission deadline** for ICLR 2024 is on **September 28**. According to the paper submission principles, this is a paper of the same period and does not require comparison.
> 2. To make the experiment more convincing, we **fairly compared MDE and COT** under the same experimental setting. It can be seen that without calibrating the classifier, the performance of **MDE is significantly better than COT**.
> 3. Within such a short rebuttal period, we do not wish to engage in a dispute over which one is better. Considering that MDE and COT belong to **different technical systems** and do **not have an incremental innovation relationship**, our innovation lies in the domain of energy score systems.
>
> **W5: Limited Dataset Diversity**
>
> We follow the reviewer's recommendation to report the result on the **Wilds datasets with more natural shifts**, the experimental setting kept the same as the ATC paper:
>
> **Table 3. Mean Absolute Error (MAE) comparison with existing methods on Wilds Setup**
>
> | Dataset | Shift | ConfScore | Entropy | Frechet | ATC | COT | NuclearNorm | AvgEnergy | MDE |
> | :---: | :---: | :---: | :---: | :---: | :---: | :---: | :---: | :---: | :---: |
> | Camelyon17 | Natural Shift | 9.01 | 8.19 | 8.49 | 7.46 | 5.31 | 4.26 | 3.21 | **2.93** |
> | RxRx1 | Natural Shift | 6.63 | 6.32 | 5.49 | 5.45 | 4.45 | 3.67 | 2.86 | **1.62** |
> | FMoW | Natural Shift | 10.52 | 9.61 | 7.47 | 6.13 | 5.26 | 4.49 | 2.90 | **2.24** |
>
> **The MAE of MDE** in predicting OOD test accuracy **on various Wilds datasets is optimal** again.
>
> By the way, the Breeds dataset is created by more finely stratifying the class structure of ImageNet. Since we have completed the experiment on ImageNet, we have not reported on the Breeds dataset.

---

> ### Author Response · Authors · 2023-11-20
> **Author Response to Reviewer SNMZ (Part IV)**
>
> **Q4: Missing a related work**
>
> We will add a description of this study [9] in the related work section: *“Chen et al. (2022) propose the SEES framework to estimate performance shift by considering the joint shift of both labels and features.”*
>
> [1] Devin Guillory et al. Predicting with confidence on unseen distributions. ICCV 2021.
>
> [2] Jiefeng Chen et al. Detecting errors and estimating accuracy on unlabeled data with self-training ensembles. Neurips 2021.
>
> [3] Thomas Unterthiner et al. Predicting neural network accuracy from weights. arXiv 2020.
>
> [4] Charles H Martin et al. Predicting trends in the quality of state-of-the-art neural networks without access to training or testing data. Nature Communications 2021.
>
> [5] Weijian Deng et al. What does rotation prediction tell us about classifier accuracy under varying testing environments? ICML 2021.
>
> [6] Ru Peng et al. CAME: Contrastive automated model evaluation. ICCV 2023.
>
> [7] Weijie Tu et al. A bag-of-prototypes representation for dataset-level applications. CVPR 2023.
>
> [8] Yuzhe Lu et al. Characterizing Out-of-Distribution Error via Optimal Transport, Neurips 2023.
>
> [9] Chen et al. Estimating and explaining model performance when both covariates and labels shift. In NeurIPS 2022

---

### Official Review · Reviewer_zk65 · 2023-11-07

**Soundness:** 2 fair
**Presentation:** 2 fair
**Contribution:** 1 poor
**Rating:** 6
**Confidence:** 3

**Summary:**

- Authors propose a method for a problem called 'AutoEval'. The problem is described as evaluating the effectiveness of a model on data without ground truth labels.
- The proposed method is very simple. It is based on energy models.
- Authors perform experiments on datasets like CIFAR-10/100, TinyImagenet to validate their method.

**Strengths:**

The authors have performed a wide range of analysis experiments.

**Weaknesses:**

1. The problem addressed
-  Authors motivate some new problem called Automated model evaluation. The definition is not very clear as they define it in many different ways.
- I am not sure about the relevance of the problem. Or to phrase it better - I don't know much about this problem.

2. Proposed method
- The proposed method just use energy based model equation to create a simple function of the energy expressed in terms of logits.
- One concern here is that it might be very similar to some method in Uncertanity estimation.
- Can authors mathematically compare this method to some common methods in uncertainity estimation.

3. Experimental validation
- While the introduction had motivated the problem in a broad setting. Authors discussed OOD to motivate the problem. But I could not find the experiments on OOD.
- The claims should match the experiments on which the method is validated. Maybe the authors can provide experiments on OOD.


------
authors have addressed the questions I asked above. So I have updated to weak accept. Weak accept because experiments are still not on very large datasets

**Questions:**

Please see the weakness section

---

> ### Author Response · Authors · 2023-11-20
> **Author Response to Reviewer zk65 (Part I)**
>
> We would like to thanks for your valuable feedback and far-sighted insights. We hope the following statements will provide you gain an in-depth understanding of AutoEval and our MDE method, thereby dispelling your concerns.
>
> **Q1:  A clear restatement of AutoEval Problem**
>
> **i. What is AutoEval?**
>
> AutoEval task is to evaluate the performance of a trained model on datasets without accessing the real labels.
>
> Specifically, given a trained model $f$ and an unlabeled OOD dataset $D_u=\lbrace x_i \rbrace_{i=1}^N$  with $N$ samples, AutoEval aims to predict the accuracy of $f$ on $D_u$.
>
> The common AutoEval workflow is to develop an indicator that is correlated to accuracy but to calculate without requiring labels.
>
> The AutoEval method is assessed by the correlation coefficient between the indicator and the accuracy, as well as the MAE between the predicted accuracy and ground-truth accuracy on OOD datasets.
>
>  **ii. Why is AutoEval needed?**
>
> 1. The AutoEval task is long-standing and has been studied in a wide range of areas, e.g. text classification [1], structured data [2], active learning [3], database [4], medicine images [5], self-driving images [6], AIGC [7], LLM [9,9], etc.
>
> 2. The statistics for these AutoEval papers appearing in the top-tier publications: **Nature Communications**(1, [10]),  ICML(9), NeurIPS(6), ICLR(4), JMLR(2), AISTATS(1), CVPR(4, including the **CVPR2020 best paper nominations** [11]), ICCV(2), ACL(1), EMNLP(2), ECAL(1), SIGMOD(1), ICSE(1), TNNLS(1), a total of **36** paper.
>
> 3. Also, **reviewers *Dsfd* and *2CiW* both agreed** in Strengths 1 of their reviews that **AutoEval is valuable** in the real-world deployment of DNNs.
>
> **Q2: How does AutoEval differ from uncertainty estimation?**
>
>   Unlike uncertainty estimation to estimate the confidence of model predictions to convey information about when a model’s output should (or should not) be trusted, AutoEval directly predicts the accuracy of model output.
>
>   In this regard, AutoEval is a task that assesses the effectiveness and deployment worthiness of a model by directly predicting its accuracy in a testing environment.
>
> **Q3: Unlike common methods in uncertainty estimation, MDE is not an incremental approach.**
>
> **i. Using logits is a general operation:**
>
> Notably, AutoEval methods based on model outputs (e.g. ConfScore [12], Entropy [13], ATC [14], NuclearNorm [15], etc.) all utilize logit, because it is closely associated with model accuracy.
>
> So MDE just adopted this conventional practice and has achieved remarkable performance.
>
> **ii. MDE are totally different from common methods in uncertainty estimation**
>
> Since the reviewer did not clarify whether MDE is similar to which methods in uncertainty estimation, we compare MDE with the widely used Softmax confidence:
>
>   1. Comparing the essence of the two methods, MDE is the meta-distribution statistic of non-normalized energy at the dataset level, while Softmax confidence is the maximum value of the normalized logit vector for a single sample.
>
> 2. Comparing the mathematical form of the two methods, they are completely different:
>
>     \begin{equation}
>     \mathcal{MDE}(x;f) =-\frac{1}{|N|} \sum_{i=1}^N \log \operatorname{Softmax} E(x;f) =-\frac{1}{|N|} \sum_{i=1}^N \log
>     \frac{e^{E\left(x_n;f\right)}}{\sum_{i=1}^N e^{E\left(x_i;f\right)}},
>     \end{equation}
>
>     \begin{equation}
>     \max _y p(y \mid \mathbf{x})=\max _y \frac{e^{f_y(\mathbf{x})}}{\sum_i e^{f_i(\mathbf{x})}}=\frac{e^{f^{\max }(\mathbf{x})}}{\sum_i e^{f_i(\mathbf{x})}}
>     \end{equation}
>
> 3. Compare the two methods for accuracy prediction on OOD data, we decompose the softmax confidence by logarithmizing it as follows:
>     \begin{equation}
>     \begin{aligned}
>     \log \max _y p(y \mid \mathbf{x}) & =E\left(\mathbf{x} ; f(\mathbf{x})-f^{\max }(\mathbf{x})\right) \\\\
>     & =E(\mathbf{x} ; f)+f^{\max }(\mathbf{x})
>     \end{aligned}
>     \end{equation}
>
>     Then, we find $f^{max}(x)$ tends to be lower and $E(x; f)$ tends to be higher for OOD data, and vice versa. This shifting results in Softmax confidence that is no longer suitable for accuracy prediction, while MDE is not affected by this bothersome issue.
>
> **iii. Reviewer *2CiW* also acknowledged the relatively high novelty of the MDE method** in Strengths 2 of his/her review.

---

> ### Author Response · Authors · 2023-11-20
> **Author Response to Reviewer zk65 (Part II)**
>
> **Q4:  Please provide experiments on OOD**
>
> - **If the reviewer is referring to "AutoEval problem inspired by the large variation in model accuracy on the ood dataset, but we did not provide the experiments on ood datasets"**:
>
>     **Actually, all experiments in the paper were conducted on OOD.**
>     We assume that this misunderstanding arises from the lack of explicit indication in the captions of the experimental
>     figures/tables that the dataset is actually OOD.
>
>     The correlations for each dataset in Table 1 are actually computed on their corresponding synthetic sets, involving the application of synthetic shift to the source dataset.
>     For instance, the correlation for CIFAR-10 specifically refers to the $\rho$, $R^2$ calculated on CIFAR-10-C.
>
>     In Table 4, we calculate the MAE on an unseen test set with natural shifts (i.e., an OOD dataset) in comparison to the source data.
>     For instance, for the CIFAR-10 dataset, we predict the model‘s accuracy on the CINIC-10 OOD dataset.
>
> - **If the reviewer intends to convey that "OOD detection has inspired the AutoEval problem, but we did not provide experiments on ood detection”**:
>
>     **i.** AutoEval and OOD detection **cannot be directly compared** as they are two **entirely different problems**. Among them, OOD detection is to **identify outlier test samples** that are different from training distributions, but AutoEval is towards **unsupervised estimating of the model’s testing accuracy.**
>
>     **ii.** This paper **was not inspired by OOD detection** techniques, but rather by the characteristics of the energy that fulfilled our desire to build an efficient and effective AutoEval framework.
>
> [1] Mayee Chen et al. Mandoline: Model evaluation under distribution shift. ICML 2021.
>
> [2] Simona Maggio et al. Performance prediction under dataset shift. ICPR 2022.
>
> [3] Yoo, Donggeun et al. Learning loss for active learning. CVPR 2019.
>
> [4] Schelter, Sebastian et al. Learning to validate the predictions of black box classifiers on unseen data. SIGMOD 2020.
>
> [5] Emmanouil Antonios Platanios et al. Estimating accuracy from unlabeled data: A bayesian approach. ICML 2016.
>
> [6] Licong Guan et al. Instance segmentation model evaluation and rapid deployment for autonomous driving using domain differences. IEEE Transactions on Intelligent Transportation Systems,  2023.
>
> [7] Yujie Lu et al. Llmscore: Unveiling the power of large language models in text-to-image synthesis evaluation. Neurips 2023.
>
> [8] Yue X et al. Automatic evaluation of attribution by large language models. EMNLP 2023 Findings.
>
> [9] Kuhn et. Semantic uncertainty: Linguistic invariances for uncertainty estimation in natural language generation. ICLR 2023.
>
> [10] C. H. Martin. Predicting trends in the quality of state-of-the-art neural networks without access to training or testing data. Nature Communications, 2021.
>
> [11] Ciprian A Corneanu et al. Computing the testing error without a testing set. CVPR, 2021.
>
> [12] Dan Hendrycks et al. A baseline for detecting misclassified and out-of-distribution examples in neural networks. ICLR 2017.
>
> [13] Devin Guillory et al. Predicting with confidence on unseen distributions. CVPR 2021.
>
> [14] Saurabh Garg et al. Leveraging unlabeled data to predict out-of-distribution performance. ICLR 2022.
>
> [15] Weijian Deng et al. Characterizingterising prediction matrix for unsupervised accuracy estimation. ICML 2023.

---

> > ### Comment · Reviewer_zk65 · 2023-12-01
> > **Response to authors**
> >
> > The authors have thoroughly addressed my concerns, therefore I am updating my rating to weak accept.

---

### Official Review · Reviewer_xmk2 · 2023-11-08

**Soundness:** 2 fair
**Presentation:** 3 good
**Contribution:** 2 fair
**Rating:** 6
**Confidence:** 3

**Summary:**

This paper intends to solve the auto evaluation problem of a well-trained classification model on a test dataset with domain shift and without labels. The proposed method is based on an energy-based framework to estimate the Meta-Distribution Energy, which is used to train a regression model on synthesized dataset for prediction of classification accuracy on unlabelled test data.

**Strengths:**

This paper introduces an energy-based automatic evaluation framework designed to enhance efficiency and mitigate overconfidence in existing methodologies. The proposed approach indicates better prediction on unseen test data over the other measurement methods on dataset in different modalities and with different classification models.

**Weaknesses:**

1. It is recommended to use a different symbol for the normalization term E(x) in Eq(2) to avoid confusion, like Z(x).
2. In Eq(5) and Eq(6), the font of matchcal is usually used for a single letter.
3. I cannot see a clear relationship between the proposed method and the energy-based model except that "energy" specifies the logits from the classification model.
4. The performance of the proposed framework relies on the regression between MDE on the synthesized dataset and its accuracy. However, the type of domain shift is more complicated in the real world in most cases, thus very unpredictable.
5. The experiment on adjusting temperature T should be conducted on a broader range, like from 0.01 to 100. The change from 1 to 10 is relatively small.
6. A real dataset for evaluation needs to be included. The operation of shear, equalization, and color temperature adjustment is easy to synthesize, while the domain shift could come from more complex sources like [1][2].
7. Some missing EBM references can be considered to be included into this paragraph to make it more complete for the readers. For example, [3] is the first EBM using CNN for energy function and trains it with Langevin dynamics. [3] is also the first one to point out that EBM and a classifier can be derived from each other. The EBM applications not only include video as you have mentioned in your paper, but also include point cloud [4], voxel [5], trajectory [6] and molecules [7].

[1] Scanner invariant multiple sclerosis lesion segmentation from MRI. 2020 IEEE 17th International Symposium on Biomedical Imaging (ISBI). IEEE, 2020.

[2] Transfer learning for domain adaptation in MRI: Application in brain lesion segmentation. Medical Image Computing and Computer Assisted Intervention− MICCAI 2017: 20th International Conference, Quebec City, QC, Canada, September 11-13, 2017, Proceedings, Part III 20. Springer International Publishing, 2017.

[3] A theory of generative convnet. ICML 2016.

[4] Generative PointNet: Deep Energy-Based Learning on Unordered Point Sets for 3D Generation, Reconstruction and Classification.

[5] Learning Descriptor Networks for 3D Shape Synthesis and Analysis. CVPR 2018.

[6] Energy-Based Continuous Inverse Optimal Control.

[7] Molecular Graph Generation with Energy-Based Models.

**Questions:**

1. For Eq(5), is MDE defined on specific data $x_n$, where MDE(x; f) should be MDE($x_n$, f)? Or does it miss an expectation term over the dataset?
2. How is the synthesized test data generated when training the regression model for accuracy prediction?
3. If the proposed method indicates a significant drop in a new dataset, is there any way to correct this bias based on the proposed MDE?
4. How is the correlation evaluated in Table 1 on a specific dataset?
5. How to determine the best hyper-parameter of T on a new dataset? Is the best parameter related to the dataset or the classification model?
6. Will the selection of different regression models affect the prediction accuracy?
7. Is there any comparison on the evaluation of domain-shift data with real labels?

---

> ### Author Response · Authors · 2023-11-20
> **Author Response to Reviewer xmk2 (Part I)**
>
> We would like to thanks for your thoughtful and insightful reply! We sincerely hope our response can resolve your concerns.
>
> **W3: Unclear relationship between MDE and EBM.**
>
> A standard Energy-Based Model (EBM) is trained with implicit MCMC sampling [1,2] to learn an energy function that assigns low energy values to input x in the sampled data distribution.
>
> **The proposed MDE is not an EBM**, but an indicator calculated with resort to the energy function of EBM. Importantly, the MDE indicator can predict the model performance on unlabeled OOD datasets in real-world scenarios, achieving the SOTA benchmark without requiring much extra overhead.
>
> In the original paper, we first introduce that the classifier can be regarded as an EBM (Eq.1~Eq. 3), and then we calculate the MDE indicator through the "energy" of all data points. (Eq.4).
>
> That is the relationship between MDE and EBM.
>
> **W4&W6: Transformations such as shear etc. is easy to synthesize, and requires more complicated real dataset for evaluation**
>
> As suggested by the reviewer, to substantiate the feasibility of MDE in the wild, we **compare MDE with other strong baselines on the more complicated real datasets — ImageNet and Wilds,** where Wilds includes the MRI-like medical image dataset Camelyon17.
>
> **Table 1. Correlation comparison with existing methods on ImageNet Setup**
>
> | Dataset | Network | ConfScore |  | Entropy |  | Frechet |  | ATC |  | COT |  | NuclearNorm |  | AvgEnergy |  | MDE |  |
> | :---: | :---: | :---: | :---: | :---: | :---: | :---: | :---: | :---: | :---: | :---: | :---: | :---: | :---: | :---: | :---: | :---: | :---: |
> |  |  |  $ρ$ | $r^2$ |  $ρ$ | $r^2$ |  $ρ$ | $r^2$ |  $ρ$ | $r^2$ |  $ρ$ | $r^2$ |  $ρ$ | $r^2$ |  $ρ$ | $r^2$ |  $ρ$ | $r^2$ |
> | ImageNet | ResNet-152 | 0.980 | 0.949 | 0.979 | 0.946 | 0.945 | 0.879 | 0.980 | 0.899 | 0.968 | 0.943 | 0.979 | 0.961 | 0.980 | 0.955 | **0.982** | **0.967** |
> |  | DenseNet-169 | 0.983 | 0.953 | 0.981 | 0.931 | 0.942 | 0.878 | 0.982 | 0.891 | 0.963 | 0.934 | 0.981 | 0.956 | 0.984 | 0.955 | **0.985** | **0.960** |
> |  | VGG-19 | 0.966 | 0.933 | 0.968 | 0.909 | 0.978 | 0.910 | 0.975 | 0.886 | 0.981 | 0.926 | 0.978 | 0.949 | 0.980 | 0.950 | **0.987** | **0.952** |
> |  | Average | 0.976 | 0.945 | 0.976 | 0.929 | 0.955 | 0.889 | 0.979 | 0.892 | 0.971 | 0.934 | 0.979 | 0.955 | 0.981 | 0.953 | **0.985** | **0.960** |
>
> **Table 2. Mean Absolute Error (MAE) comparison with existing methods on ImageNet Setup**
>
> | Dataset | Unseen test set | ConfScore | Entropy | Frechet | ATC | COT | NuclearNorm | AvgEnergy | MDE |
> | :---: | :---: | :---: | :---: | :---: | :---: | :---: | :---: | :---: | :---: |
> | ImageNet | ImageNet-V2-A | 8.76 | 9.72 | 6.36 | 6.88 | 7.04 | 4.04 | 3.58 | **2.21** |
> |  | ImageNet-V2-B | 8.59 | 10.20 | 9.51 | 8.82 | 9.02 | 5.20 | 4.41 | **2.35** |
> |  | ImageNet-V2-C | 14.92 | 10.27 | 9.12 | 8.74 | 8.85 | 5.66 | 5.01 | **4.27** |
> |  | ImageNet-S | 7.30 | 9.50 | 10.45 | 9.81 | 8.11 | 7.77 | 6.40 | **5.31** |
> |  | ImageNet-R | 15.41 | 13.63 | 12.06 | 12.70 | 12.87 | 11.34 | 10.64 | **7.68** |
> |  | ImageNet-Vid | 15.18 | 14.26 | 16.53 | 14.42 | 13.49 | 12.77 | 11.19 | **8.09** |
> |  | ImageNet-Adv | 19.09 | 18.28 | 18.37 | 19.20 | 14.51 | 13.61 | 11.43 | **8.42** |
> |  | Average | 12.75 | 12.27 | 11.77 | 11.51 | 10.56 | 8.63 | 7.52 | **5.48** |
>
> **Table 3. Mean Absolute Error (MAE) comparison with existing methods on Wilds Setup**
>
> | Dataset | Shift | ConfScore | Entropy | Frechet | ATC | COT | NuclearNorm | AvgEnergy | MDE |
> | :---: | :---: | :---: | :---: | :---: | :---: | :---: | :---: | :---: | :---: |
> | Camelyon17 | Natural Shift | 9.01 | 8.19 | 8.49 | 7.46 | 5.31 | 4.26 | 3.21 | **2.93** |
> | RxRx1 | Natural Shift | 6.63 | 6.32 | 5.49 | 5.45 | 4.45 | 3.67 | 2.86 | **1.62** |
> | FMoW | Natural Shift | 10.52 | 9.61 | 7.47 | 6.13 | 5.26 | 4.49 | 2.90 | **2.24** |
>
>
> **The considerable performance on both ImageNet and Wilds reflects the versatility of MDE**. In the future, we envision deploying the MDE into broader real-world domains to demonstrate its scalability.
>
> [1] Yilun Du et al. Implicit generation and modeling with energy based models. Neurips 2019.
>
> [2] Will Grathwohl, et al. Your classifier is secretly an energy based model and you should treat it like one. ICLR 2020.

---

> ### Author Response · Authors · 2023-11-20
> **Author Response to Reviewer xmk2 (Part II)**
>
> **W5&Q5: Small range temperature ablation and how to determine the best temperature**
>
> The ablation results for a wide range of temperature hyperparameters **($T$ from 0.1 to 100)** are shown below:
>
> | T | 0.01 | 0.5 | 1 | 2 | 3 | 4 | 5 | 6 | 7 | 8 | 9 | 10 | 20 | 30 | 40 | 50 | 60 | 70 | 80 | 90 | 100 |
> | --- | --- | --- | --- | --- | --- | --- | --- | --- | --- | --- | --- | --- | --- | --- | --- | --- | --- | --- | --- | --- | --- |
> |  $\rho$ | 0.988 | 0.989 | 0.991 | 0.991 | 0.990 | 0.988 | 0.987 | 0.986 | 0.984 | 0.983 | 0.983 | 0.982 | 0.976 | 0.974 | 0.974 | 0.974 | 0.973 | 0.971 | 0.972 | 0.971 | 0.971 |
> | $r$ | 0.989 | 0.988 | 0.987 | 0.981 | 0.974 | 0.968 | 0.964 | 0.961 | 0.960 | 0.959 | 0.959 | 0.958 | 0.959 | 0.959 | 0.959 | 0.960 | 0.960 | 0.960 | 0.960 | 0.960 | 0.961 |
> | $r^2$ | 0.978 | 0.977 | 0.974 | 0.963 | 0.948 | 0.937 | 0.929 | 0.924 | 0.921 | 0.920 | 0.919 | 0.918 | 0.919 | 0.920 | 0.921 | 0.921 | 0.921 | 0.921 | 0.922 | 0.922 | 0.923 |
> | MAE | 1.72 | 1.80 | 1.78 | 1.94 | 2.28 | 2.71 | 3.32 | 4.25 | 5.10 | 6.32 | 7.98 | 9.50 | 9.87 | 10.05 | 10.16 | 10.22 | 10.24 | 10.25 | 10.25 | 10.24 | 10.26 |
>
> In fact, despite the ideal case for MDE being $T \rightarrow 0$ in Theorem 3.1, our empirical results found that the performance variation caused by $T$ from 0 to 1 is minimal.
> Thus, we fix $T = 1$ to ensure that the classification accuracy is not affected so that we can compare fairly with baselines. Most importantly, we did **not rely on T to tune performance**.
>
> **W1&W2&W7: Non-standard symbol format and missing EBM citations.**
>
> Thanks for the reminders! We will complete these symbols and references.
>
> **Q1: A clerical error about MDE.**
>
> Sorry, we missed a summation sign after the $-\frac{1}{|N|}$ please excuse this clerical error.
>
> Eq.(5) is an expectation term：$\mathcal{MDE}(x;f) =-\frac{1}{|N|} \sum_{i=1}^N \log \operatorname{Softmax} E(x;f) =-\frac{1}{|N|} \sum_{i=1}^N \log \frac{e^{E\left(x_n;f\right)}}{\sum_{i=1}^N e^{E\left(x_i;f\right)}}.$
>
> **Q2&Q4: How to generate the synthesized set for regression and evaluate correlation in Table 1?**
>
> We take the **CIFAR-10 setup** as an example to clarify this doubt. At this setup, **the synthetic set is CIFAR-10-C datasets**, which involve 19 types of corruption with 5 different intensity levels applied to the CIFAR-10 validation set.
> When **encountering a new data setup**, we can perform **the same corruption operation on its validation set** to obtain the synthetic set.
> **The correlation score** of each dataset in Table 1 refers to **the coefficients $\rho$,  $R^2$ calculated on the <MDE, accuracy> pair** of its corresponding **synthesized set**.
>
> **Q3: How to correct performance drop on new datasets？**
>
> - **_If the reviewer wants to know “how to mitigate the MAE and correlation degradation of the MDE method on OOD datasets”_**:
>
>     **All AutoEval methods may suffer from performance degradation on unknown new datasets**, and this is not exclusive to MDE alone.
>
>     This performance decline typically arises from severe domain shifts between the new dataset and the training dataset, such as adversarial perturbations and extreme class imbalances.
>     Faced with these challenging scenes, how to sample the new dataset to capture its main shift types and combine it into the synthetic set may be a promising solution.
>     We sincerely request the community to actively involve themselves in addressing this issue.
>
> - **_If the reviewer wants to know “how to alleviate the accuracy decline of models on OOD datasets”_**:
>
>     Once MDE finds that the model performance drops significantly on the new dataset, we can introduce related **technologies of domain adaptation or domain generalization** to alleviate this problem, but this is not the scope of this article.
>
> **Q6: Comparing the performance of different regression models.**
>
> The choice of different regression models **does not affect the accuracy prediction performance**. With a fixed dataset (CIFAR-10) and backbone (VGG11), we compare the correlation and MAE results between RobustLinearRegression and LinearRegression below:
>
> |  | RobustLinearRegression | LinearRegression |
> | :---: | :---: | :---: |
> |  $\rho$ | 0.991 | 0.991 |
> | $r$ | 0.987 | 0.987 |
> | $r^2$ | 0.973 | 0.974 |
> | MAE | 1.80 | 1.78 |

---

> > ### Comment · Reviewer_xmk2 · 2023-11-22
> >
> > Is the pdf file updated with the latest revision?

---

> > > ### Author Response · Authors · 2023-11-22
> > >
> > > We would like to express our heartfelt gratitude for your prompt response.
> > >
> > > First and foremost, we sincerely apologize for not being able to synchronize all the updated content into the PDF version of the paper in a timely manner.
> > >
> > > Due to the relatively substantial contents we need to prepare for discussion, unfortunately, we ran out of time to update the PDF file.
> > >
> > > We will make every effort to promptly supplement all the updates into the **_Section Appendix_** before the disscusiion deadline.

---

> ### Author Response · Authors · 2023-11-20
> **Author Response to Reviewer xmk2 (Part III)**
>
> **Q7: Is there any comparison on the evaluation of domain-shift data with real labels?**
>
> **Yes, we have conducted all experiments of this work on the domain-shift dataset.**
> We assume that this misunderstanding arises from the lack of explicit indication in the captions of the experimental figures/tables that the dataset is actually OOD.
>
> The correlation for each dataset in Table 1 is actually computed on its corresponding **synthetic sets**, which **is a synthetic domain-shifted dataset** for the source dataset,
>
> In Table 4, we calculate the MAE on the **unseen test sets**, which **are natural domain-shifted datasets** in comparison to the source dataset. For instance, for the CIFAR-10 dataset, we evaluate the correlation on the CIFAR-10-C synthetic domain-shifted dataset and calculate the MAE on natural domain-shifted datasets such as CINIC-10, etc.
>
> Notably, the MDE method does not require access to the real labels when predicting the model accuracy on domain-shift data.
> If we understand the reviewer right, we do need the real labels to provide real accuracy when evaluating the MAE of the MDE method.

---

### Author Response · Authors · 2023-11-22
**Sincerely Looking Forward to the Reviewers' Feedback**

Dear reviewers:

We sincerely appreciate your careful evaluation and thoughtful feedback. We hold the utmost respect for your diligent efforts.

As the review discussion stage is drawing to a close, we humbly seek to ascertain whether our responses adequately address your concerns. We greatly appreciate your attention to this matter.

We have summarized our responses to each reviewer as follows:

1. We clarified the definition, significance, and relationship to uncertainty estimation, OOD detection, and energy-based models in regard to the AutoEval problem. (To Reviewers **xmk2, zk65**)
2. We elaborated on the advantages (compared to prior work), contributions, and theoretical grounding of MDE. (To Reviewers **zk65, SNMZ**)
3. We expanded experiments on complicated real-world datasets of interest to the reviewers, such as ImageNet and Wilds Datasets, along with strong baselines like COT.  (To Reviewers **xmk2, SNMZ**)
4. We added experiments to address the reviewer's inquiries regarding the temperature hyperparameter range and the selection of regression models. (To Reviewers **xmk2, 2CiW**)
5. We explained the experimental setup, such as synthetic dataset generation, correlation evaluation, and experiments on OOD data. (To Reviewers **xmk2, zk65**)
6. We provided additional analysis on storage cost and time-consuming. (To Reviewers **SNMZ, Dsfd, 2CiW**)
7. We will complete all citations and formulas in the next draft as requested. (To Reviewers **xmk2, SNMZ**)

If you have any additional questions or require further clarification, please do not hesitate to inform us. We value the opportunity to continue the discussion if necessary.

Best regards,

The authors

---

### Author Response · Authors · 2023-11-23
**All responses have been updated to the latest PDF version**

Dear reviewers,

We genuinely appreciate the depth of your insightful comments and the value of your feedback. Your dedicated efforts are truly esteemed.

**We have incorporated responses to all reviewers into the latest PDF version, with each modification highlighted in blue across the whole paper.**

Due to the deadline is approaching, we sincerely hope you can take a look at it. We faithfully hope our response adequately address your concern.

Best regards,

The authors

---

### Meta-Review · Area_Chair_Aq1g · 2023-12-07

**Metareview:**

The paper proposes a novel measure, Meta-Distribution Energy (MDE), enhancing the efficiency and effectiveness of the AutoEval framework. MDE establishes a meta-distribution statistic on the information (energy) associated with individual samples, providing a smoother representation enabled by energy-based learning. Extensive experiments across modalities, datasets, and different architectural backbones validate MDE's validity and demonstrate its superiority over prior approaches. The paper's novelty lies in introducing energy scores into the AutoEval field for the first time, setting it apart from existing methods. This research is valuable as it addresses the challenge of assessing actual performance in practical applications. The extensive experiments sufficiently validate the model's performance, and the theoretical analysis also provides additional justification. The Area Chair recommends accepting the paper and encourages the authors to revise it by incorporating the valuable suggestions from the reviewers.

**Justification For Why Not Higher Score:**

Despite receiving support from most reviewers and demonstrating value, the Area Chair thinks that the paper is not strong enough to be accepted as a spotlight because the novelty of the method is marginally above average.

**Justification For Why Not Lower Score:**

The paper introduces a novel method that serves as a valuable tool for addressing the challenge of assessing actual performance in practical applications. It provides extensive experimental results and theoretical explanations, justifying the proposed method and making a comprehensive contribution to the field.

---

### Decision · Program_Chairs · 2024-01-16

Accept (poster)